# Consequences of telomere dysfunction in fibroblasts, club and basal cells for lung fibrosis development

Sergio Piñeiro-Hermida[1], Paula Martínez[1,4], Giuseppe Bosso[1,4], Juana María Flores [2], Sarita Saraswati[1], Jane Connor[3], Raphael Lemaire[3] & Maria A. Blasco [1] ✉

TRF1 is an essential component of the telomeric protective complex or shelterin. We previously showed that dysfunctional telomeres in alveolar type II (ATII) cells lead to interstitial lung fibrosis. Here, we study the lung pathologies upon telomere dysfunction in fibroblasts, club and basal cells. TRF1 deficiency in lung fibroblasts, club and basal cells induced telomeric damage, proliferative defects, cell cycle arrest and apoptosis. While *Trf1* deletion in fibroblasts does not spontaneously lead to lung pathologies, upon bleomycin challenge exacerbates lung fibrosis. Unlike in females, *Trf1* deletion in club and basal cells from male mice resulted in lung inflammation and airway remodeling. Here, we show that depletion of TRF1 in fibroblasts, Club and basal cells does not lead to interstitial lung fibrosis, underscoring ATII cells as the relevant cell type for the origin of interstitial fibrosis. Our findings contribute to a better understanding of proper telomere protection in lung tissue homeostasis.

Telomeres are heterochromatic structures at the chromosome ends, which are essential for chromosome stability. In mammals, telomeric DNA consists of TTAGGG tandem repeats bound by the so-called shelterin complex, which encompasses TRF1, TRF2, TIN2, POT1, TPP1 and RAP1 protein[1,2]. Shelterin complex ensures telomere protection by preventing end-to-end chromosome fusions, telomere fragility, and activation of DNA damage response[3]. With each cell division, telomeres shorten due to the incomplete replication of chromosome ends[4,5]. Telomere shortening can be compensated through the de novo addition of telomeric repeats by telomerase, a reverse transcriptase composed of a catalytic subunit (TERT) and an RNA component (Terc)[6].

Specifically, TRF1 has a relevant role in shelterin complex assembly[7–9]. TRF1 is also important to prevent telomere fusions, telomeric DNA damage and multitelomeric signals, as well as for the replication of telomeric DNA[10,11]. Moreover, TRF1 was reported to be essential for the induction and maintenance of pluripotency[12,13].

Conditional deletion of *Trf1* in specific cell types has demonstrated and important role in tissue regeneration and homeostasis[10,13–15].

Idiopathic pulmonary fibrosis (IPF) is a fibrosing interstitial lung disease characterized by the histopathological pattern of usual interstitial pneumonia. IPF affects 3 million people worldwide with a median survival time from diagnosis of 2–4 years[16–18]. Interestingly, between 8% and 15% of familial IPF cases are associated with mutations in telomerase or in telomere-protective proteins[19–24]. Interestingly, sporadic cases of IPF, not associated with telomerase mutations, also show shorter telomeres compared to age-matched controls, with 10% of the patients showing telomeres as short as the telomerase mutation carriers[19]. Telomerase mutations have also been found in up to 1% of smokers showing chronic obstructive pulmonary disease (COPD), also leading to abnormally short telomeres[25].

Lung fibroblasts are mesenchymal cells of the interstitium with a key role in the formation and extension of alveolar septa and in alveolar epithelial proliferation and differentiation. After alveolar or

[1]Telomeres and Telomerase Group, Molecular Oncology Program, Spanish National Cancer Centre (CNIO), Melchor Fernández Almagro 3, Madrid E-28029, Spain. [2]Animal Surgery and Medicine Department, Faculty of Veterinary Science, Complutense University of Madrid, Madrid, Spain. [3]Research and Early Development, Respiratory, Inflammation and Autoimmune (RIA), BioPharmaceuticals R&D, AstraZeneca, Gaithersburg, MD 20878, USA. [4]These authors contributed equally: Paula Martínez, Giuseppe Bosso. ✉e-mail: mblasco@cnio.es

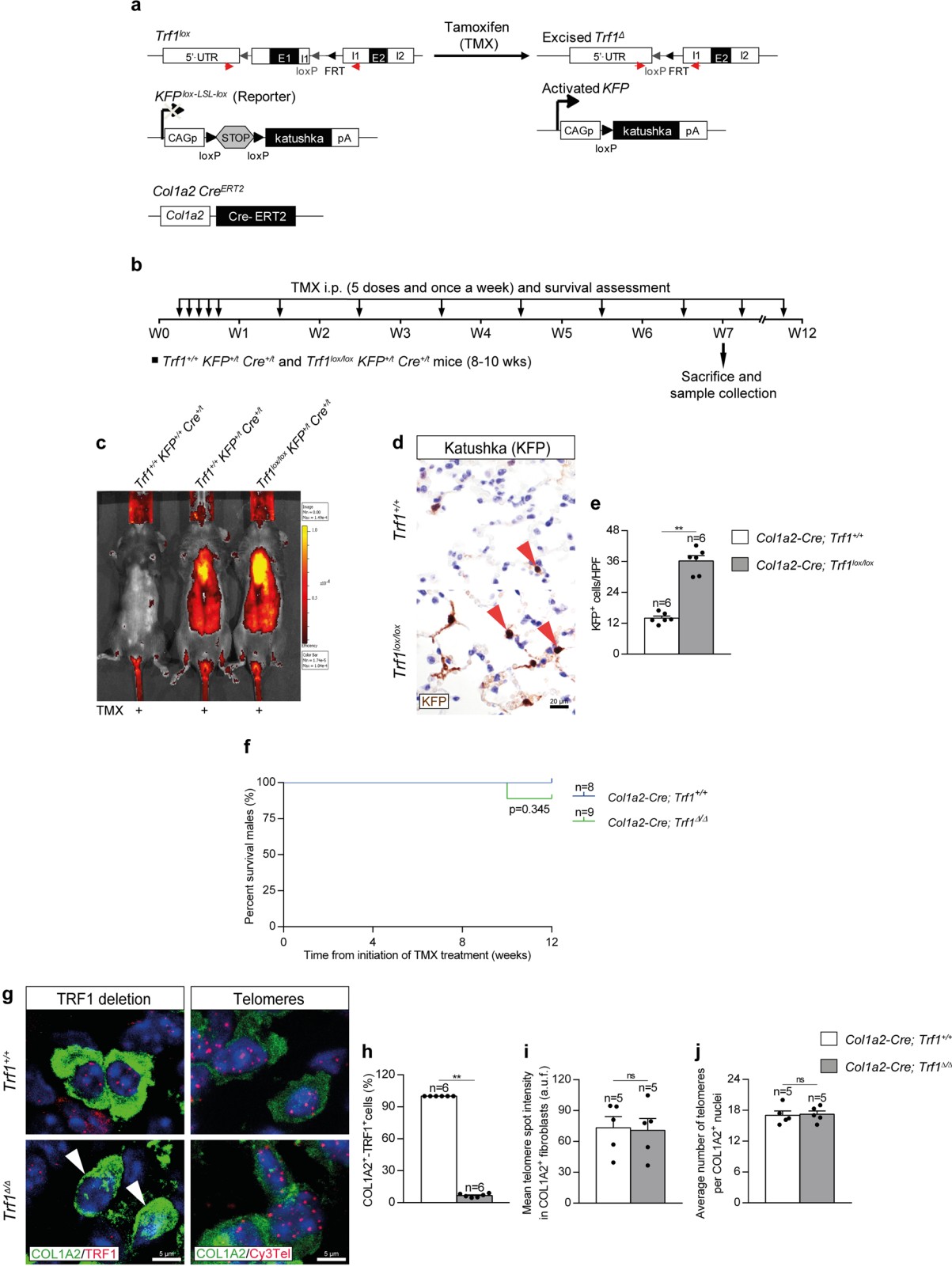

endothelial-cell injury or immune activation and inflammation, fibroblasts can be activated and proliferate and differentiate into myofibroblasts, which further contribute to pulmonary fibrosis[26,27]. On the other hand, alveolar type II (ATII) cells are localized in the gaseous alveolar surfaces, and their main function is the secretion of surfactant to prevent alveolar collapse. ATII cells have been reported as progenitor cells for ATI cells, having a key role in lung repair after injury[28–30]. Club cells are airway epithelial cells whose main roles are detoxification of xenobiotic and oxidizing molecules, secretion of antimicrobial peptides, and promotion of mucociliary clearance. Club cells are primary progenitor cells, since the play a key role in bronchiolar epithelial repair through their ability to self-renew and differentiate into ciliated and goblet cells[31–34]. Basal cells are only present in the mouse trachea, whereas in humans, they extend to the respiratory

**Fig. 1 | Efficient *Trf1* deletion in lung fibroblasts upon tamoxifen administration. a** Generation of the conditional knockout mouse model in which *Trf1* was deleted in fibroblasts using the Cre recombinase driven by the *Col1a2* promoter. *Trf1^lox^*, *KFP^Lox-LSL-Lox^*, and *Col1a2-Cre^ERT2^* alleles are depicted before and after Cre-mediated excision. **b** Tamoxifen (TMX) treatment, survival rate assessment and sample collection. Eight-to 10-week-old male *Trf1^+/+^ KFP^+/t^ Cre^+/t^* (*Col1a2-Cre; Trf1^+/+^*) and *Trf1^lox/lox^ KFP^+/t^ Cre^+/t^* (*Col1a2-Cre; Trf1^lox/lox^*) mice were i.p. injected with TMX for five consecutive days during the first week and once a week until the sacrifice and sample collection on week (W) 7, and during the follow-up of survival until W12. **c** Representative images of fluorescence intensity of katushka fluorescent protein (KFP) in *Trf1^+/+^ KFP^+/+^ Cre^+/t^*, *Trf1^+/+^ KFP^+/t^ Cre^+/t^* and *Trf1^lox/lox^ KFP^+/t^ Cre^+/t^* mice. Representative immunostainings for KFP (**d**), and quantification of KFP positive cells per 40X high-power field (HPF) (**e**) in lung sections from *Trf1^+/+^ KFP^+/+^ Cre^+/t^* and *Trf1^lox/lox^*

*KFP^+/t^ Cre^+/t^* mice. **f** Kaplan−Meier survival curves of *Col1a2-Cre; Trf1^+/+^* (*Trf1^+/+^*, controls) and *Col1a2-Cre; Trf1^Δ/Δ^* (*Trf1^Δ/Δ^*) mice upon TMX treatment. **g** Representative immunofluorescence stainings for COL1A2 (green) and TRF1 (red) (white arrowheads indicate COL1A2^+^ fibroblasts with deletion of TRF1), and immune-telomere-Q-FISH in COL1A2^+^ fibroblasts (Cy3Tel probe (red), COL1A2^+^ cells (green), and nuclei stained with DAPI (blue)) in lung sections from *Trf1^+/+^* and *Trf1^Δ/Δ^* mice. Quantification of the proportion of double COL1A2^+^-TRF1^+^ fibroblasts (**h**) and mean telomere spot intensity (**i**) and average number of telomeres (**j**) in COL1A2^+^ cells from *Trf1^+/+^* and *Trf1^Δ/Δ^* mice. Data are expressed as mean ± SEM (the number of mice is indicated in each case). $**p < 0.01$ (Mann–Whitney or unpaired $t$ tests). Animal survival was assessed by the Kaplan–Meier analysis, using the log Rank (Mantel–Cox) test. Source data are provided as a Source Data file.

bronchioles. In a steady state basal cells are quiescent; however, in response to injury, airway basal cells become activated and operate as stem/progenitor cells capable of self-renewal and differentiation into ciliated and secretory cells[33,35,36].

Noteworthy, we have shown that induction of telomere dysfunction in ATII cells is sufficient to induce progressive and lethal pulmonary fibrosis in mice[15]. However, lung pathological consequences of telomere dysfunction in fibroblasts, club and basal cells have not yet been investigated. On this basis, we have conditionally deleted in mice the sheltering component *Trf1* in fibroblasts, club and basal cells to examine the pathological consequences of dysfunctional telomeres on the lung. Here, we show that depletion of TRF1 in fibroblasts, Club and basal cells does not lead to interstitial fibrosis, underscoring ATII cells as the relevant cell type for the origin of interstitial lung fibrosis.

## Results

### Efficient Trf1 deletion in lung fibroblasts upon tamoxifen administration

We induced a short-term telomere dysfunction in fibroblasts by genetically deleting the shelterin component *Trf1* in these cells, which have been reported to contribute to pulmonary fibrosis[26,27]. In particular, to specifically delete *Trf1* in lung fibroblasts we generated a *Trf1^lox/lox^ KFP^Lox-LSL-Lox^ Col1a2-Cre^ERT2^* mouse model, in which Cre recombinase expression is controlled by the collagen, type I, alpha 2 (*Col1a2*) promoter, which is specific for fibroblasts[37,38]. CreERT2 is conditionally activated by tamoxifen (TMX) administration[37]. Moreover, a transgene encoding for the katushka fluorescent protein (KFP) that contains a stop cassette flanked by lox sequences was introduced as a reporter to monitor Cre activity[15] (Fig. 1a). TMX was administered intraperitoneally (i.p.) for five consecutive days during the first week and once a week until the sacrifice and sample collection on week (W) 7, and during the follow-up of survival until W12 (Fig. 1b). First, we assessed the fluorescence intensity of KFP in *Trf1^+/+^ KFP^+/+^ Cre^+/t^*, *Trf1^+/+^ KFP^+/t^ Cre^+/t^* (*Col1a2-Cre; Trf1^+/+^*) and *Trf1^lox/lox^ KFP^+/t^ Cre^+/t^* (*Col1a2-Cre; Trf1^lox/lox^*) mice by in vivo fluorescence imaging. As expected, KFP fluorescence was solely detected in *KFP^+/t^* mice. Fluorescence intensity was apparently higher in the lungs of *Col1a2-Cre; Trf1^lox/lox^* mice compared to *Col1a2-Cre; Trf1^+/+^* (controls) mice after 7 weeks of TMX treatment (Fig. 1c). Accordingly, the amount of KFP-positive cells per 40X high-power field (HPF) evaluated by immunohistochemistry was higher in the lungs of *Col1a2-Cre; Trf1^lox/lox^* mice compared to control mice (Fig. 1d, e). We did not find differences in survival between *Col1a2-Cre; Trf1^+/+^* (*Trf1^+/+^*, controls) and *Col1a2-Cre; Trf1^Δ/Δ^* (*Trf1^Δ/Δ^*) mice for as long as 12 weeks of TMX treatment (Fig. 1f). Immunofluorescence staining with antibodies against COL1A2 and TRF1 clearly demonstrated that the majority of COL1A2-positive cells stained negative for TRF1 (93%) in the lungs of *Trf1^Δ/Δ^* mice (Fig. 1g, h). Moreover, we performed an immune-telomere-Q-FISH for the quantification of mean telomere spot intensity and average number of telomeres in COL1A2-positive cells, which are readouts of telomere length[10]. The results did not show any significant difference either in telomere fluorescence or in the numbers of

detectable telomeres between wild-type and *Trf1^Δ/Δ^* fibroblasts, indicating that *Trf1* deletion in lung fibroblasts does not result in telomere length changes (Fig. 1g, i, j).

### Trf1 deletion in lung fibroblasts increases telomeric damage, cell cycle arrest and apoptosis, and reduces proliferation of fibroblasts

In order to study the effects of TRF1 deficiency on lung fibroblasts, we performed double immunostainings of COL1A2 with markers of DNA damage (γH2AX), cell cycle arrest (p16 and p21), apoptosis (C3) and proliferation (Ki67), as well as an an immuno-telomere-Q-FISH with the DNA damage marker 53BP1 to identify telomeric induced foci (TIF) in COL1A2 positive cells in lung sections from *Trf1^Δ/Δ^* mice and controls (Fig. 2a−g). *Trf1^Δ/Δ^* mice exhibited increased numbers of lung fibroblasts with DNA damage, cell cycle arrest, apoptosis and proliferation, as well as increased proportion of COL1A2^+^ cells with more than 1 TIF (Fig. 2a−g).

Next, to evaluate the potential lung pathological effects of *Trf1* deletion in lung fibroblasts, we assessed cellularity and total protein concentration in bronchoalveolar lavage fluid (BALF), mRNA expression of Th1 inflammation markers, Sirius Red staining (collagen deposition), immunostaining for Vimentin (fibroblast presence), as well as lung function by plethysmography (lung resistance (LR) and dynamic compliance (Cdyn)) in our mouse cohorts (Fig. 2h−t). We found elevated BALF cell counts for neutrophils and lymphocytes in *Trf1^Δ/Δ^* as compared to wild-type mice (Fig. 2h−l). Nevertheless, lung mRNA expression of Th1 inflammation markers *Tnf*, *Il1b*, *Il6* and *Ifn-γ* (Fig. 2m−p), as well as alveolar Sirius Red and Vimentin-stained areas (Fig. 2q, r) were not found significantly changed between experimental groups, indicating that the short-term deletion of *Trf1* specifically in fibroblasts does not result in either inflammation nor fibrosis. In agreement, lung function was not altered as judged by similar lung resistance (LR) and dynamic compliance (Cdyn) in both genotypes (Fig. 2s, t).

### Trf1 deletion in lung fibroblasts exacerbates profibrotic pathologies upon bleomycin-induced pulmonary fibrosis

As we did not see significant lung pathologies in *Col1a2* mutants, we set to study the effect of TRF1 deficiency in fibroblasts in the context of a bleomycin (BLM)-induced fibrosis model. For this purpose, eight-to 10-week-old male *Trf1^+/+^ KFP^+/t^ Cre^+/t^* (*Col1a2-Cre; Trf1^+/+^*) and *Trf1^lox/lox^ KFP^+/t^ Cre^+/t^* (*Col1a2-Cre; Trf1^lox/lox^*) mice were intraperitoneally (i.p.) administered with TMX for five consecutive days during the first week and then once a week until week (W) 7 to induce the deletion of *Trf1* in *Col1a2^+^* fibroblasts (Fig. 3a). Then, at W7, animals were intra-tracheally instilled with either a single dose of 0.8 mg/kg of BLM or saline solution (Fig. 3a). First, we performed an immunofluorescence staining with antibodies against COL1A2 and TRF1 that clearly demonstrated that the majority of COL1A2-positive cells stained negative for TRF1 (88%) in the lungs of *Trf1^Δ/Δ^* mice. Upon BLM challenge, only the 33 % of COL1A2-positive cells stained negative for TRF1 at the end-point three weeks post-BLM (Fig. 3b, c).

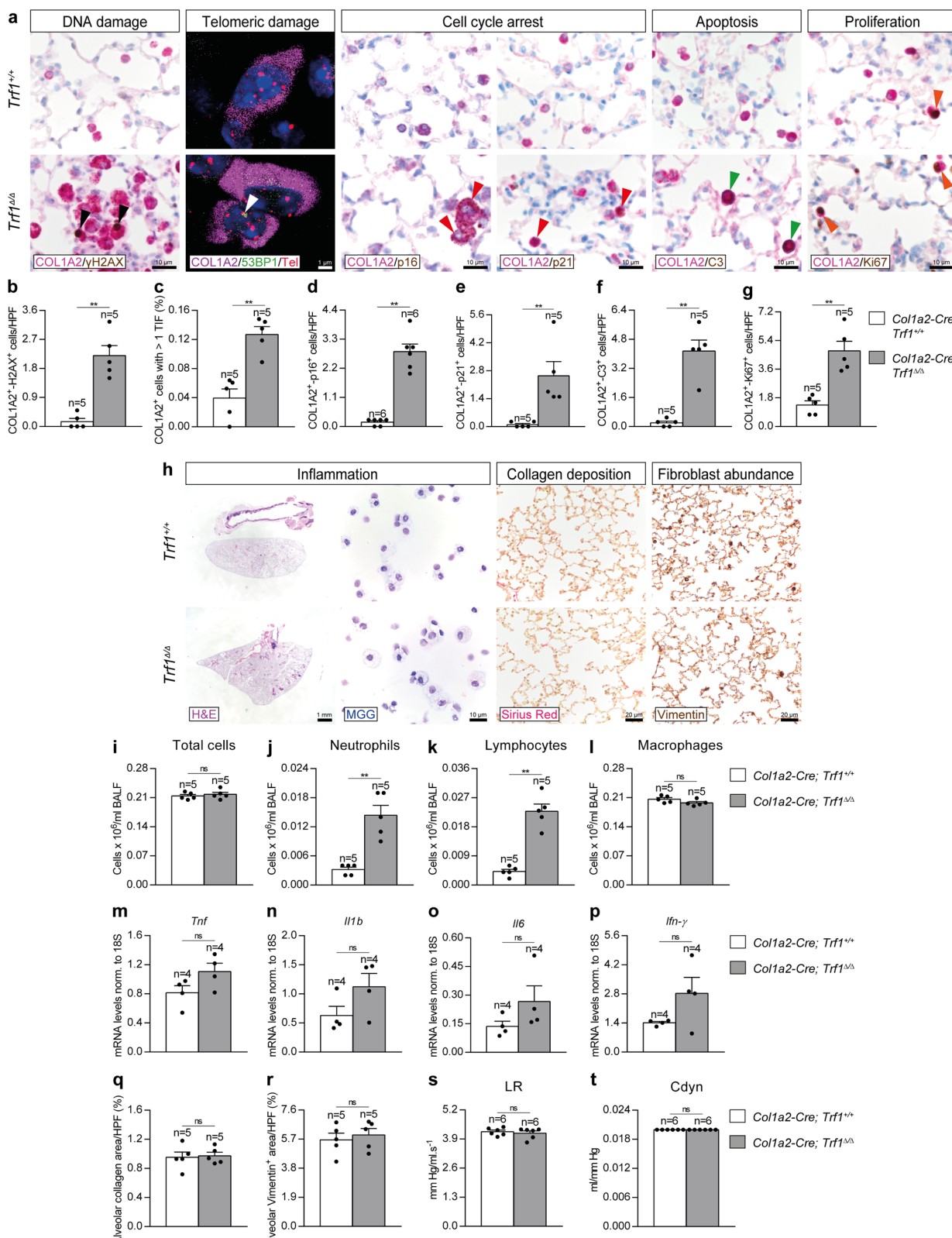

In order to assess collagen deposition we performed Sirius Red stainings in our mouse cohorts. Collagen deposition was found significantly increased after BLM challenge, being this increment more pronounced in *Trf1*^Δ/Δ mice (Fig. 3b, d). As we observed that *Trf1* deletion in fibroblasts increased collagen deposition in the lungs of *Trf1*^Δ/Δ mice after BLM treatment, we next studied the expression of several lung profibrotic markers. To this end, we performed

immunostainings for Vimentin and Smooth Muscle Actin (SMA) (fibroblast abundance) and E-Cadherin (Epithelial-Mesenchymal Transition, EMT), as well as double immunostainings of SMA with the proliferation marker Ki67 in lung sections of our mouse cohorts (Fig. 3b, e–h). Of note, Vimentin, SMA and E-Cadherin positive areas, as well as the number of SMA proliferating cells were found significantly incremented after BLM challenge, being this increase more

**Fig. 2 | *Trf1* deletion in lung fibroblasts increases telomeric damage, cell cycle arrest and apoptosis, and reduces proliferation of lung fibroblasts.**
**a** Representative lung immunostainings for COL1A2 (purple) and γH2AX (brown; black arrowheads indicate double COL1A2⁺-γH2AX⁺ fibroblasts), COL1A2 (purple) and p16 and p21 (brown; red arrowheads indicate double COL1A2⁺-p16⁺ and COL1A2⁺-p21⁺ fibroblasts), COL1A2 (purple) and C3 (brown; green arrowheads indicate double COL1A2⁺-C3⁺ fibroblasts), and COL1A2 (purple) and Ki67 (brown; orange arrowheads indicate double COL1A2⁺-Ki67⁺ fibroblasts), as well as representative lung images of telomeric induced foci (TIF) in COL1A2⁺ cells (COL1A2 (purple), Cy3Tel probe (red), 53BP1⁺ cells (green; white arrowheads indicate TIF) and nuclei stained with DAPI (blue)) in lung sections from *Trf1*⁺/⁺ and *Trf1*^Δ/Δ mice. Quantification of COL1A2⁺-γH2AX⁺ (**b**), COL1A2⁺-p16⁺ (**d**), COL1A2⁺-p21⁺ (**e**), COL1A2⁺-C3⁺ (**f**), and COL1A2⁺-Ki67⁺ (**g**) fibroblasts per 40X high-power field (HPF), as well as the proportion (%) of COL1A2⁺ cells with more than 1 TIF in lung sections from *Trf1*⁺/⁺ and *Trf1*^Δ/Δ mice (**c**). **h** Representative images of *Trf1*⁺/⁺ and *Trf1*^Δ/Δ lungs (H&E), BALF cytospin preparations (May-Grünwald Giemsa (MGG)) and Sirius Red staining, and Vimentin immunostainings in lung sections from *Trf1*⁺/⁺ and *Trf1*^Δ/Δ mice. Quantification of total (**i**) and differential BALF cell counts for neutrophils (**j**), lymphocytes (**k**) and macrophages (**l**), and Lung tissue mRNA expression levels of *Tnf* (**m**) *Il1b* (**n**), Il6 (**o**) and *Ifn-γ* (**p**) (Th1 inflammation) in *Trf1*⁺/⁺ and *Trf1*^Δ/Δ mice. Quantification of alveolar collagen (Sirius Red) (**q**) and Vimentin (**r**) positive areas (%), and lung resistance (LR) (**s**) and dynamic compliance (Cdyn) (**t**) evaluated by plethysmography in *Trf1*⁺/⁺ and *Trf1*^Δ/Δ mice. Data are expressed as mean ± SEM (the number of mice is indicated in each case). **$p < 0.01$ (Mann–Whitney or unpaired *t* tests). Source data are provided as a Source Data file.

pronounced in *Trf1*^Δ/Δ mice (Fig. 3b, e–h). We next assessed lung tissue mRNA expression of *Ccl12* (recruitment of fibrocytes), *Col1a1*, *Col1a2*, *Col3a1*, *Col4a1*, *Col5a1* and *Col6a1* (collagen markers), as well as an ELISA to quantify TGFB1 protein levels on lung homogenates (Fig. 3i–p). In agreement with increased fibroblast abundance (Fig. 3b, e, f), these markers were found elevated upon BLM treatment, being this increase more pronounced in *Trf1*^Δ/Δ mice (Fig. 3i–p).

To better evaluate the potential lung pathological effects of *Trf1* deletion in lung fibroblasts upon BLM treatment, we assessed lung function by plethysmography (Fig. 3q, r). We observed that lung resistance (LR) was increased after BLM challenge in *Trf1*⁺/⁺ and *Trf1*^Δ/Δ mice compared to controls but this increment was higher in *Trf1*^Δ/Δ mice (Fig. 3q). Of note, dynamic compliance (Cdyn) was only found significantly decreased in *Trf1*^Δ/Δ mice after BLM treatment (Fig. 3r).

**Trf1 deletion in lung fibroblasts exacerbates bleomycin-induced inflammatory response**
To evaluate the effect of *Trf1* deficiency in fibroblasts on lung inflammation, we assessed cellularity and total protein concentration in bronchoalveolar lavage fluid (BALF), as well as performed immunostainings for the quantification of neutrophils (MPO), T lymphocytes (CD4 and CD8) and macrophages (F4/80) in lung sections from our mouse cohorts (Fig. 4a–k). Specifically, neutrophil and lymphocyte counts in BALF were found elevated in *Trf1*^Δ/Δ control mice as compared to *Trf1*⁺/⁺ mice. Of note, BLM treatment resulted in increased total and differential cell counts, being this increase significantly higher in *Trf1*^Δ/Δ mice in the case of total cells and neutrophils (Fig. 4a–f). Moreover, BLM challenge increased total protein concentration in BALF, being this increment more pronounced in *Trf1*^Δ/Δ mice (Fig. 4g). Accordingly, *Trf1*^Δ/Δ control lungs exhibit increased presence of MPO⁺ neutrophils, CD4⁺ and CD8⁺ lymphocytes and F4/80⁺ macrophages compared to *Trf1*⁺/⁺ mice. BLM treatment induced an increase in the presence of these immune cell types along with F4/80⁺ macrophages that was more pronounced in *Trf1*^Δ/Δ mice (Fig. 4h–k).

We next quantified lung tissue mRNA expression of Th1 (*Tnf*, *Il1b*, *Il6* and *Ifng*) and Th2 (*Il4*, *Il10* and *Il13*) inflammation markers (Fig. 4l–r). BLM challenge increased the expression of Th1 and Th2 inflammation markers, and this increment was more pronounced in *Trf1*^Δ/Δ mice in the case of *Il6*, *Ifng*, *Il4*, *Il10* and *Il13* (Fig. 4l–r). These results clearly show that deletion of *Trf1* in fibroblasts exacerbates the BLM-induced inflammatory response.

**Efficient Trf1 deletion in club cells upon tamoxifen administration**
Next, we induced telomere dysfunction in club cells by genetically deleting *Trf1* in these cells, which constitute a lung stem cell population that plays a key role in bronchiolar epithelial repair through their ability to self-renew and differentiate into ciliated and goblet cells[31,32]. To delete *Trf1* in club cells, we generated the *Trf1*^lox/lox *KFP*^Lox-LSL-Lox *Scgb1a1-Cre*^ERT2 mouse model, in which Cre recombinase expression is controlled by the secretoglobin, family 1A, member 1 (*Scgb1a1*) promoter, which is specific for club cells (Fig. 5a). CRE-ERT2 is conditionally activated by TMX administration[31]. In addition, a transgene encoding for the KFP that contains a stop cassette flanked by lox sequences was introduced as a reporter to monitor Cre activity[15] (Fig. 5a). TMX was administered i.p. for five consecutive days during the first week and once a week until the end-point. Mice were sacrificed at W22 post-tamoxifen treatment (Fig. 5b). We did not find significant differences in survival between *Scgb1a1-Cre; Trf1*⁺/⁺ (*Trf1*⁺/⁺, controls) and *Scgb1a1-Cre; Trf1*^Δ/Δ (*Trf1*^Δ/Δ) male and female mice (Fig. 5c and Supplementary Fig. 1a). Of note, immunostaining with antibodies against SCGB1A1 and TRF1 clearly demonstrated that the majority of SCGB1A1-positive club cells stained negative for TRF1 (97%) in the lungs of *Trf1*^Δ/Δ male and female mice (Fig. 5d, e and Supplementary Fig. 1b). To measure telomere length specifically in club cells, we performed an immune-telomere-Q-FISH for the quantification of mean telomere spot intensity and average number of telomeres in SCGB1A1-positive cells. No differences neither in mean telomere length nor in the number of telomeric spots were found between male wild-type and *Trf1*^Δ/Δ club cells, indicating that TRF1 depletion does not affect telomere length homeostasis in this cell type (Fig. 5d, f, g).

In order to evaluate the effect of *Trf1* deficiency in club cells on peripheral blood, we evaluated total and differential white blood cell counts in male and female mice. We only found a slight increment in differential blood cell counts for neutrophils and eosinophils in *Trf1*^Δ/Δ male mice (Fig. 5h–l and Supplementary Fig. 1c–g).

**Trf1 deletion in club cells increases telomeric damage, cell cycle arrest and differentiation, and reduces proliferation of club cells**
In order to evaluate the effects of *Trf1* deficiency on club cells, we assessed the total number of club and basal cells, the numbers of club cells positive for γH2AX, p21, Ki67 and SOX2 as a read out for DNA damage, cell cycle arrest, proliferation and differentiation, respectively, as well as an immuno-telomere-Q-FISH with the DNA damage marker 53BP1 to identify telomeric induced foci (TIF) in SCGB1A1 positive cells in lung sections from *Trf1*^Δ/Δ mice and controls (Fig. 6a–h). First, we performed double immunostainings with the club and basal cell markers SCGB1A1 and p63, and with SCGB1A1 and γH2AX, p21, Ki67 and SOX2 (Fig. 6a–d, f–h) in *Trf1*⁺/⁺ and *Trf1*^Δ/Δ male mice. The results show that *Trf1*^Δ/Δ mice exhibited a reduced number of SCGB1A1 positive club cells and conversely increased presence of p63 positive basal cells per epithelium length in distal airways (Fig. 6a–c). Moreover, *Trf1*^Δ/Δ mice also showed increased number of γH2AX, p21 and SOX2 positive club cells, as well as decreased number of proliferating Ki67 positive club cells per epithelium length (Fig. 6a, d, f–h). *Trf1*^Δ/Δ mice also showed increased proportion of SCGB1A1⁺ cells with more than 1 TIF (Fig. 6a, e). Lung resistance (LR) and dynamic compliance (Cdyn) were addressed by plethysmography in *Trf1*^Δ/Δ mice and controls. Notably, only *Trf1*^Δ/Δ male mice showed a slight but significant increase in LR (Fig. 6i, j and Supplementary Fig. 1h, i).

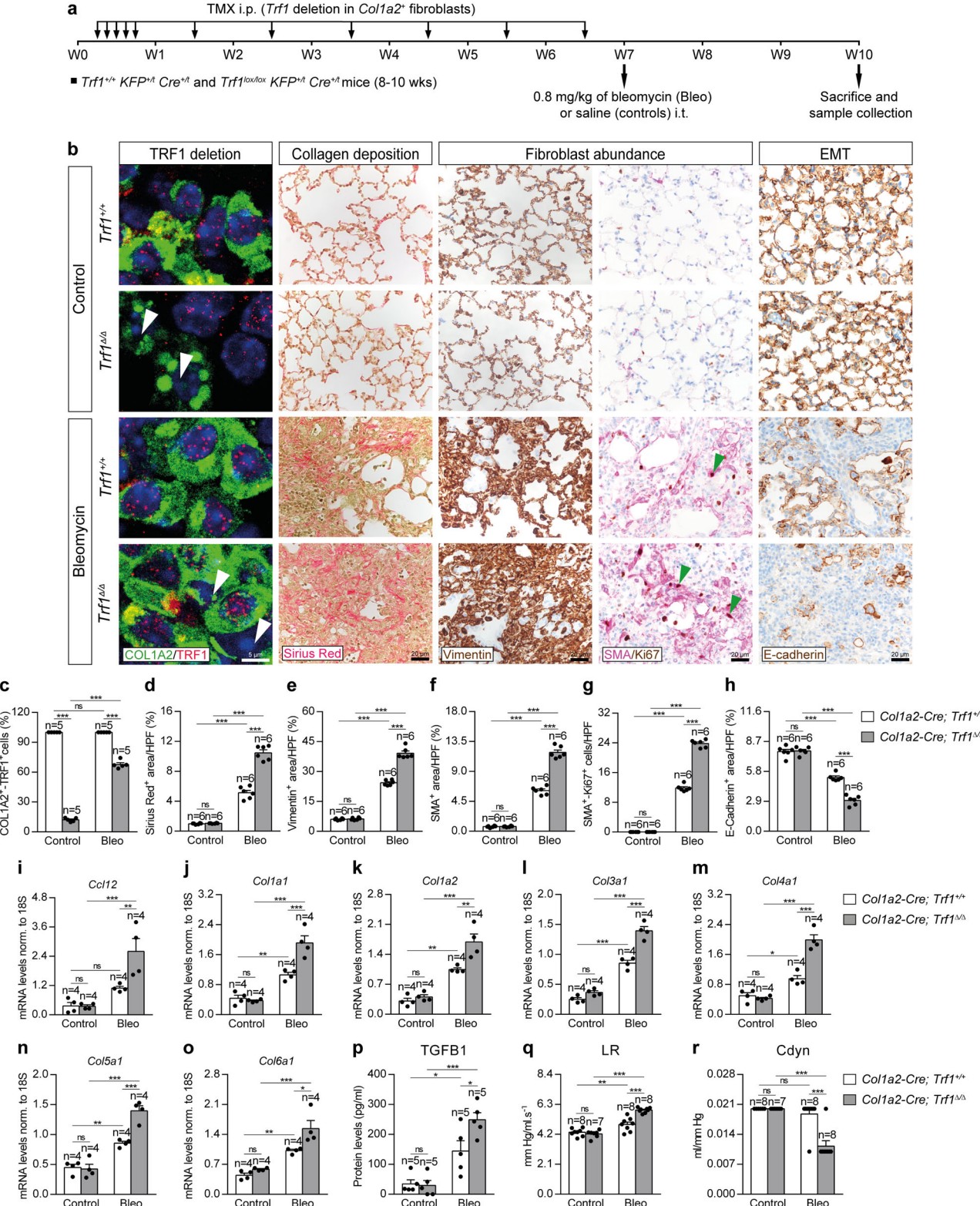

**Trf1 deletion in club cells increases lung inflammation and airway remodeling**

To study the pathological consequences of *Trf1* deficiency in club cells on the lung, we quantified cellularity and total protein concentration in BALF, as well as performed immunostainings for the quantification of neutrophils (MPO), T lymphocytes (CD4 and CD8) and macrophages (F4/80) in lung sections from *Trf1^Δ/Δ* male mice and controls, as well as lung mRNA expression of Th1 inflammation markers (Fig. 7a–n). We

found that total and differential cell counts for neutrophils, lymphocytes and macrophages as well as total protein concentration in BALF, an indicator of vascular permeability were elevated in *Trf1^Δ/Δ* as compared to wild-type control male mice (Fig. 7a–f). Accordingly, we found increased presence of neutrophils, CD4 and CD8 T lymphocytes and macrophages in lung sections from *Trf1^Δ/Δ* male mice, as well as incremented lung mRNA expression of Th1 inflammation markers *Tnf*, *Il1b*, *Il6* and *Ifn-γ* (Fig. 7g–n). These differences were not observed in

**Fig. 3 | *Trf1* deletion in lung fibroblasts exacerbates profibrotic pathologies upon bleomycin-induced pulmonary fibrosis. a** Tamoxifen (TMX) was intraperitoneally (i.p.) injected to eight-to 10-week-old male *Trf1*[+/+] *KFP*[+/t] *Cre*[+/t] (*Col1a2-Cre*; *Trf1*[+/+]) and *Trf1*[lox/lox] *KFP*[+/t] *Cre*[+/t] (*Col1a2-Cre*; *Trf1*[lox/lox]) mice for five consecutive days during the first week and then once a week until week (W) 7. Then, at W7, animals were intra-tracheally instilled with either a single dose of 0.8 mg/kg of bleomycin (BLM) or saline (controls). Sacrifice and sample collection were performed at W10. **b** Representative immunofluorescence stainings for COL1A2 (green) and TRF1 (red) (white arrowheads indicate COL1A2[+] fibroblasts with deletion of TRF1), Sirius Red stainings and immunostainings for Vimentin, Smooth Muscle Actin (SMA) and Ki67 (SMA (purple) and Ki67 (brown); green arrowheads indicate double SMA[+]·Ki67[+] fibroblasts), and E-cadherin in lung sections from control and BLM-challenged *Trf1*[+/+] and *Trf1*[Δ/Δ] mice. Quantification of the proportion of double COL1A2[+]·TRF1[+]

fibroblasts in COL1A2[+] cells (**c**), and airway collagen (Sirius Red) (**d**), Vimentin (**e**), SMA (**f**) and E-cadherin (**h**) positive areas (%), and SMA[+]·Ki67[+] fibroblasts per 40X high-power field (HPF) (**g**) in lung sections from control and BLM-challenged *Trf1*[+/+] and *Trf1*[Δ/Δ] mice. Lung tissue mRNA expression levels of *Ccl12* (recruitment of fibrocytes) (**i**), *Col1a1* (**j**), *Col1a2* (**k**), *Col3a1* (**l**), *Col4a1* (**m**), *Col5a1* (**n**) and *Col6a1* (**o**) (collagen markers), and TGFB1 (myofibroblast differentiation) protein levels (**p**) in lung homogenates from control and BLM-challenged *Trf1*[+/+] and *Trf1*[Δ/Δ] mice. Quantification of lung resistance (LR) (**q**) and dynamic compliance (Cdyn) (**r**) evaluated by plethysmography in control and BLM-challenged *Trf1*[+/+] and *Trf1*[Δ/Δ] mice. Data are expressed as mean ± SEM (the number of mice is indicated in each case). *$p < 0.05$, **$p < 0.001$, ***$p < 0.001$ (Dunn–Sidak test for multiple comparisons). Source data are provided as a Source Data file.

female mice with the exception of a slight increment in the number of neutrophils in BALF (Supplementary Fig. 1j–n).

To further investigate the consequences of *Trf1* deficiency in club cells on the lung, we performed a Sirius Red staining (collagen deposition), as well as immunostainings for Collagen I (collagen deposition), Vimentin and SMA (fibroblast abundance) in *Trf1*[Δ/Δ] male mice and controls. We did not find changes in alveolar collagen content between both experimental groups. On the other hand, airway collagen (Sirius Red), Collagen I and Vimentin stained areas, as well as airway smooth muscle (SM) thickness were found significantly increased in *Trf1*[Δ/Δ] male mice (Fig. 7o–s). Conversely, airway collagen (Sirius Red) and Vimentin-stained areas, as well as SM thickness were not significantly incremented in *Trf1*[Δ/Δ] female mice (Supplementary Fig. 1j, o, p, q). These results indicate that telomere dysfunction in club cells results in pathological pattern exclusively in the airways but not in the lung parenchyma of male mice, thus ruling out that telomere dysfunction in club cells is sufficient to induce pulmonary fibrosis in mice. Instead, these pathologies are reminiscent of airway remodeling observed in asthmatic patients, characterized by increased airway smooth muscle mass and sub-epithelial fibrosis[39].

### Efficient Trf1 deletion in lung basal cells upon tamoxifen administration

Next, we induced a telomere dysfunction in basal cells by genetically deleting *Trf1* in these cells, which in response to injury, become activated and operate as progenitor cells capable of self-renewal and differentiation into ciliated and secretory cells[33,35,36]. To delete *Trf1* in basal cells we generated the *Trf1*[lox/lox] *KFP*[Lox-LSL-Lox] *p63-Cre*[ERT2] mouse model, in which Cre recombinase expression is controlled by the transformation related protein 63 (*p63*) promoter, which is specific for basal cells CRE-ERT2 is conditionally activated by TMX administration[40]. In addition, a transgene encoding for the KFP that contains a stop cassette flanked by lox sequences was introduced as a reporter to monitor Cre activity (Fig. 8a). TMX was administered i.p. for five consecutive days during the first week and once a week until the end-point that was set at week 10 (W10) after the beginning of TMX treatment (Fig. 8b and Supplementary Fig. 2a). Following TMX administration, *p63-Cre*; *Trf1*[Δ/Δ] male and female mice exhibited a decreased survival compared to *p63-Cre*; *Trf1*[+/+] (control) mice, which was more marked in male mice (Fig. 8C and Supplementary Fig. 2a). Histopathological analysis at death point show that *Trf1*[Δ/Δ] mice exhibited abnormal pathologies in the skin, tongue, esophagus and non-glandular stomach, characterized by the presence of hyperkeratosis, dysplastic epithelial hyperplasia, dermal invasion, nuclear atypias and epithelial lamina propria invasion with inflammation (Fig. 8d). Immunostaining with antibodies against p63 and TRF1 clearly demonstrated that the majority of p63-positive cells stained negative for TRF1 (97%) in the lungs of *Trf1*[Δ/Δ] male and female mice (Fig. 8e, f and Supplementary Fig. 2b).

Quantification of mean telomere spot intensity and average number of telomeres in p63-positive cells by immune-telomere-Q-FISH

showed no significant changes in the lungs of *Trf1*[Δ/Δ] male mice as compared to control mice (Fig. 8e, g, h).

To evaluate the consequences of *Trf1* deficiency in basal cells on peripheral blood, we evaluated total and differential white blood cell counts. We only found increased total and differential blood cell counts for neutrophils and lymphocytes in *Trf1*[Δ/Δ] male mice (Fig. 8i–m) and for neutrophils and eosinophils in *Trf1*[Δ/Δ] female mice (Supplementary Fig. 2c–g).

### Trf1 deletion in basal cells increases telomeric damage and cell cycle arrest, and reduces proliferation of lung basal cells

To evaluate the effect of *Trf1* deficiency on basal cells, we assessed the total number of basal as well as double positive for p63 and either γH2AX, p21 or Ki67 as a read out for DNA damage, cell cycle arrest and proliferation, respectively, as well as an immuno-telomere-Q-FISH with the DNA damage marker 53BP1 to identify telomeric induced foci (TIF) in p63 positive cells in lung sections from in *Trf1*[Δ/Δ] male mice and controls (Fig. 9a–g). The *Trf1*[Δ/Δ] mice showed a reduced number of p63 positive basal cells per epithelium length, without significant changes in the number of club cells (Fig. 9a–c). Furthermore, *Trf1*[Δ/Δ] mice showed increased number of γH2AX and p21, as well as decreased number of Ki67 positive basal cells per epithelium length (Fig. 9a, d, f, g). *Trf1*[Δ/Δ] mice also exhibited increased proportion of p63+ cells with more than 1 TIF (Fig. 9a, e). In addition, lung resistance (LR) and dynamic compliance (Cdyn) were evaluated by plethysmography in *Trf1*[Δ/Δ] mice and controls. Of note, *Trf1*[Δ/Δ] male mice exhibited a slight increment in LR that was not observed in females, indicating that TRF1 depletion in basal cells in the airways affects male pulmonary function (Fig. 9h, i and Supplementary Fig. 2h, i).

### Trf1 deletion in lung basal cells increases lung inflammation and airway remodeling

To gain insight into how *Trf1* deficiency in basal cells affect normal lung homeostasis, we quantified cellularity and total protein concentration in BALF, as well as performed immunostainings for the quantification of neutrophils (MPO), T lymphocytes (CD4 and CD8) and macrophages (F4/80) in lung sections from *Trf1*[Δ/Δ] male mice and controls. In addition, we assessed lung mRNA expression of Th1 inflammation markers (Fig. 10a–n). We observed that total and differential BALF cell counts for neutrophils and lymphocytes were found elevated in *Trf1*[Δ/Δ] mice, with the exception of macrophages which were found significantly reduced (Fig. 10a–e). *Trf1*[Δ/Δ] female mice exhibited increased presence of neutrophils and lymphocytes as well as decreased number of macrophages in BALF (Supplementary Fig. 2j–n). Total protein concentration in BALF, an indicator of vascular permeability, was also found incremented in *Trf1*[Δ/Δ] male mice (Fig. 10f). In accordance, we found increased presence of neutrophils, CD4 and CD8 T lymphocytes and macrophages in lung sections from *Trf1*[Δ/Δ] male mice, and incremented lung mRNA expression of Th1 inflammation markers *Tnf*, *Il1b*, *Il6* and *Ifn-γ* (Fig. 10g–n).

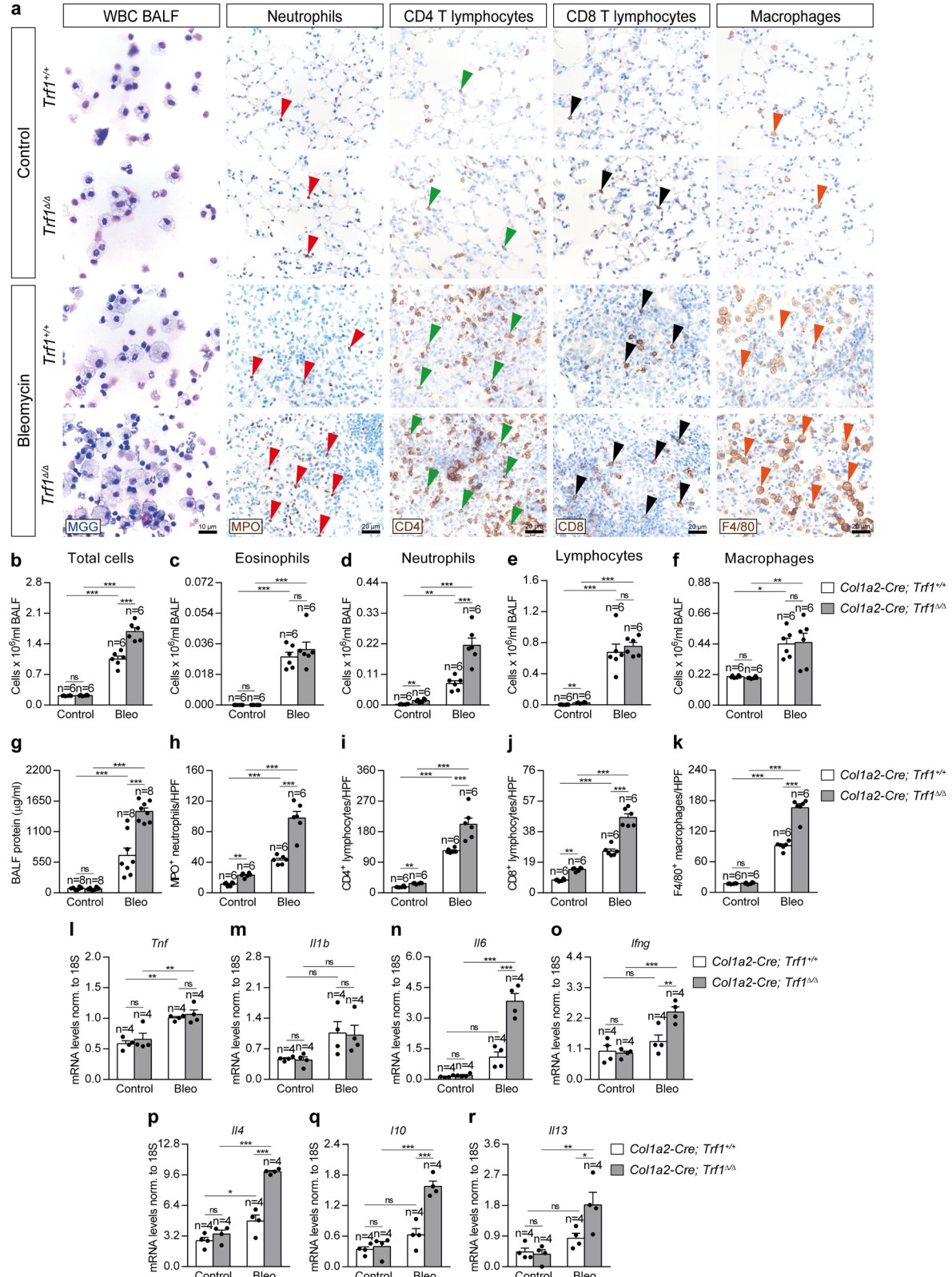

**Fig. 4 | *Trf1* deletion in lung fibroblasts exacerbates bleomycin-induced inflammatory response. a** Representative BALF cytospin preparations (May-Grün-wald Giemsa (MGG)) and immunostainings for MPO (red arrowheads indicate MPO⁺ neutrophils), CD4 and CD8 (green and black arrowheads indicate CD4⁺ and CD8⁺ T lymphocytes) and F4/80 (orange arrowheads indicate F4/80⁺ macrophages) in lung sections from control and BLM-challenged *Trf1*⁺/⁺ and *Trf1*^Δ/Δ mice. Quantification of total (**b**) and differential BALF cell counts for eosinophils (**c**), neutrophils (**d**) lymphocytes (**e**) and macrophages (**f**) and total protein concentration in BALF (**g**), of control and BLM-challenged *Trf1*⁺/⁺ and *Trf1*^Δ/Δ mice. Quantification of lung MPO (**h**), CD4 (**i**), CD8 (**j**) and F4/80 (**k**) positive cells per 40X high-power field (HPF), and lung tissue mRNA expression levels of *Tnf* (**l**) *Il1b* (**m**), *Il6* (**n**) and *Ifn-γ* (**o**) (Th1 inflammation) and *Il4* (**p**), *Il10* (**q**) and *Il13* (**r**) (Th2 inflammation) in control and BLM-challenged *Trf1*⁺/⁺ and *Trf1*^Δ/Δ mice. Data are expressed as mean ± SEM (the number of mice is indicated in each case). *$p < 0.05$, **$p < 0.01$, ***$p < 0.001$ (Dunn–Sidak test for multiple comparisons). Source data are provided as a Source Data file.

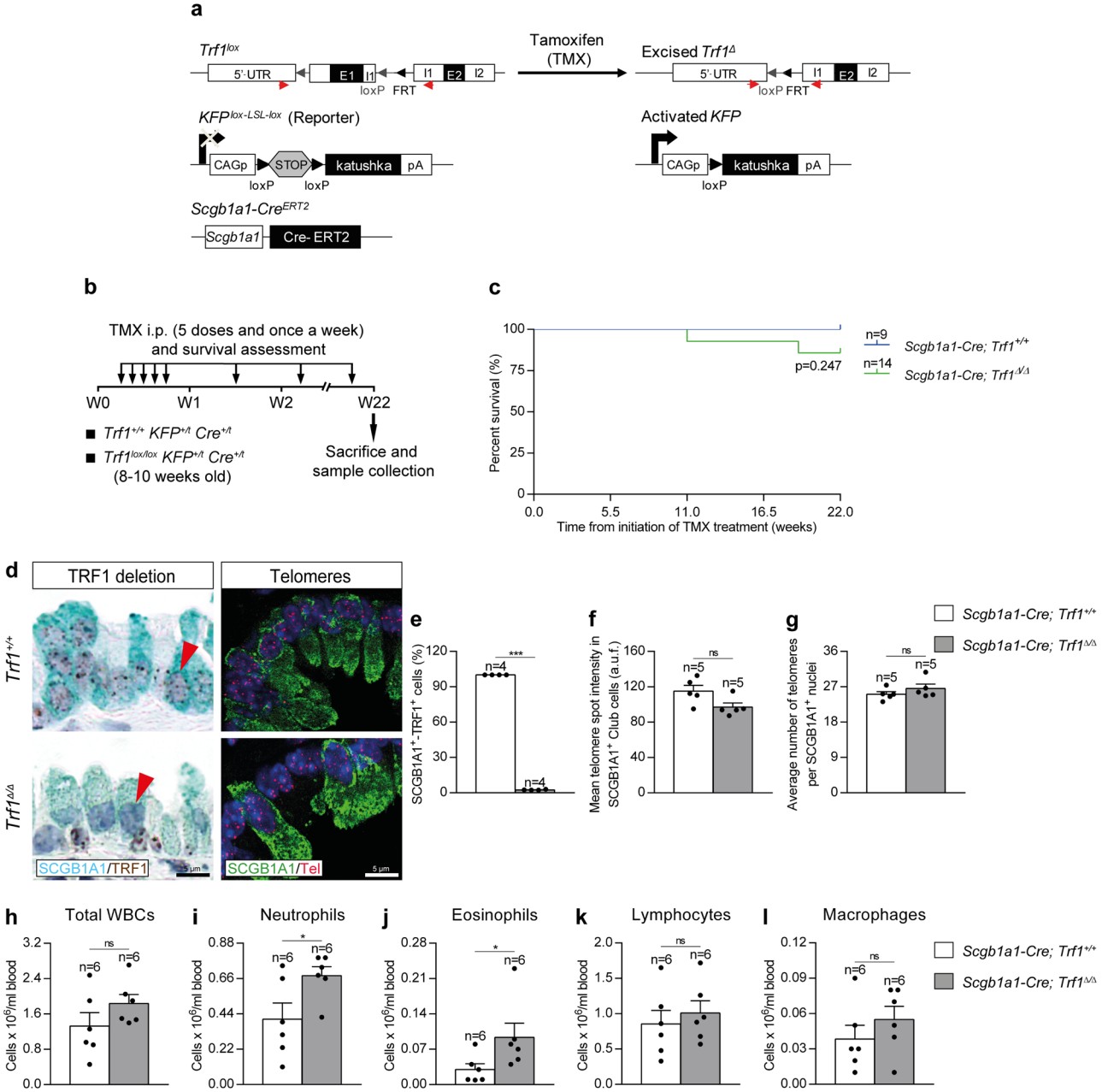

**Fig. 5 | Efficient *Trf1* deletion in club cells upon tamoxifen administration.**
**a** Generation of the conditional knockout mouse model in which *Trf1* was deleted in club cells using the Cre recombinase driven by the *Scgb1a1* promoter. *Trf1^lox^*, *KFP^Lox-LSL-Lox^*, and *Scgb1a1-Cre^ERT2^* alleles are depicted before and after Cre-mediated excision. **b** Tamoxifen (TMX) treatment, survival rate assessment and sample collection. Eight-to 10-week-old male *Trf1^+/+^ KFP^+/t^ Cre^+/t^* (*Scgb1a1-Cre; Trf1^+/+^*) and *Trf1^lox/lox^ KFP^+/t^ Cre^+/t^* (*Scgb1a1-Cre; Trf1^lox/lox^*) mice were i.p. injected with TMX for five consecutive days during the first week and once a week until the sacrifice and sample collection on week (W) 22. **c** Kaplan–Meier survival curves of *Scgb1a1-Cre; Trf1^+/+^* (*Trf1^+/+^*, controls) and *Scgb1a1-Cre; Trf1^Δ/Δ^* (*Trf1^Δ/Δ^*) mice upon TMX treatment. **d** Representative immunostainings for SCGB1A1 (blue) and TRF1 (brown) (red arrowheads indicate SCGB1A1^+^ club cells with deletion of TRF1), and immune-telomere-Q-FISH in SCGB1A1^+^ club cells (Cy3Tel probe (red), SCGB1A1^+^ cells (green), and nuclei stained with DAPI (blue)) in lung sections from *Trf1^+/+^* and *Trf1^Δ/Δ^* mice. Quantification of SCGB1A1^+^-TRF1^+^ cells (%) (**e**), mean telomere spot intensity (**f**) and average number of telomeres (**g**) in SCGB1A1^+^ cells in *Trf1^+/+^* and *Trf1^Δ/Δ^* mice. Quantification of total white blood cells (**h**), neutrophils (**i**), eosinophils (**j**), lymphocytes (**k**) and macrophages (**l**) in peripheral blood from *Trf1^+/+^* and *Trf1^Δ/Δ^* mice. Data are expressed as mean ± SEM (the number of mice is indicated in each case). *$p < 0.05$, ***$p < 0.001$ (Mann–Whitney or unpaired *t* tests). Animal survival was assessed by the Kaplan–Meier analysis, using the log Rank (Mantel–Cox) test). Source data are provided as a Source Data file.

Next, to further study the pathological consequences of *Trf1* deficiency in basal cells on the lung, we performed a Sirius Red staining (collagen deposition), as well as immunostainings for Collagen I (collagen deposition), Vimentin and SMA (fibroblast abundance) in *Trf1^Δ/Δ^* male mice and controls. We did not find changes in alveolar collagen content between experimental groups. However, in the airways there was a clear increase in collagen (Sirius Red), Collagen I and Vimentin-stained areas, as well as smooth muscle (SM) thickness in *Trf1^Δ/Δ^* male

mice (Fig. 10o–s). Of note, airway collagen (Sirius Red) and Vimentin-stained areas, as well as SM thickness were not significantly incremented in *Trf1^Δ/Δ^* female mice (Supplementary Fig. 2j, o, p, q). Again, we did not find any signs of lung parenchymal fibrosis, thus suggesting that telomere dysfunction in basal cells is not sufficient to lead to pulmonary fibrosis in mice.

Additionally, we studied the lung pathological consequences of *Trf1* deletion in basal cells when deleted from early embryonic

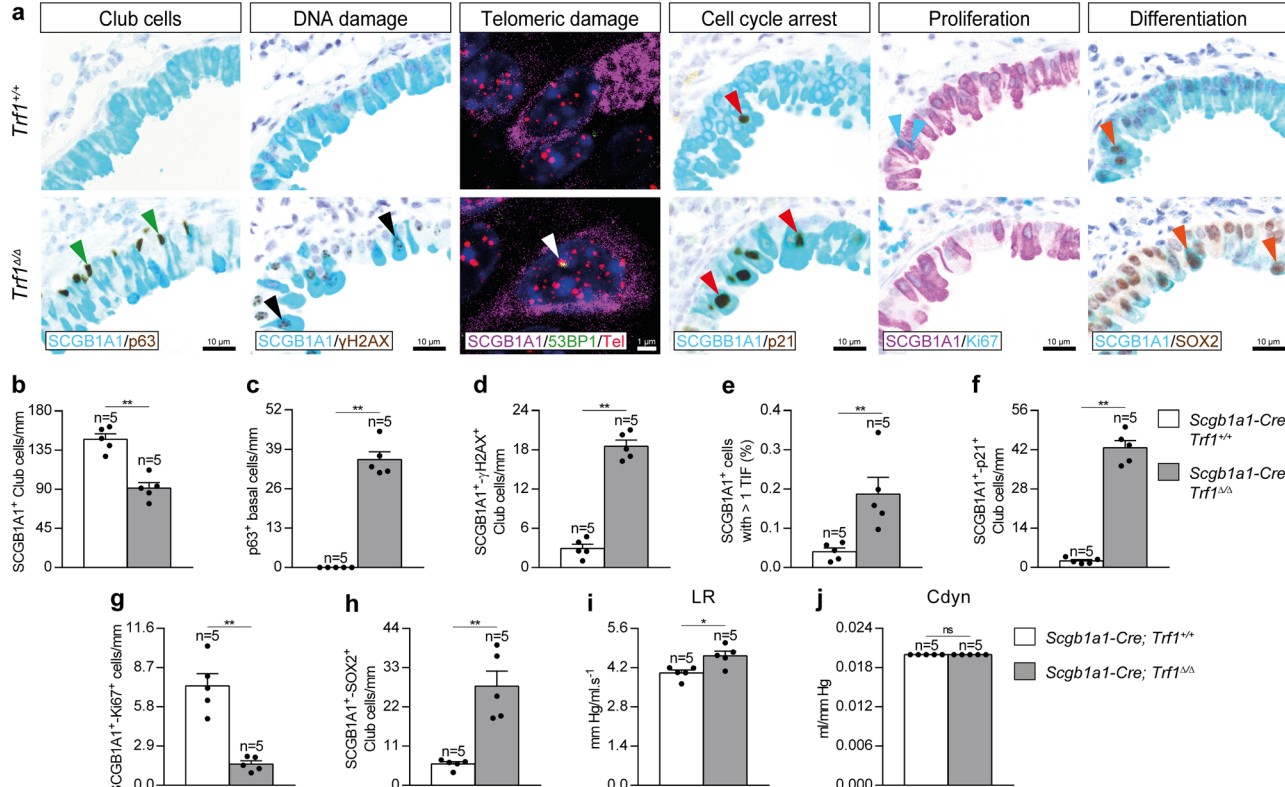

**Fig. 6 | *Trf1* deletion in club cells increases telomeric damage, cell cycle arrest and differentiation, and reduces proliferation of club cells. a** Representative immunostainings for SCGB1A1 (blue) and p63 (brown; green arrowheads indicate p63+ basal cells), SCGB1A1 (blue) and γH2AX (brown; black arrowheads indicate double SCGB1A1+-γH2AX+ club cells), SCGB1A1 (blue) and p21 (brown; red arrowheads indicate double SCGB1A1+-p21+ club cells), SCGB1A1 (purple) and Ki67 (blue; blue arrowheads indicate double SCGB1A1+-Ki67+ club cells), and SCGB1A1 (blue) and SOX2 (brown; orange arrowheads indicate double SCGB1A1+-SOX2+ club cells), as well as representative images of telomeric induced foci (TIF) in SCGB1A1+ cells (SCGB1A1 (purple), Cy3Tel probe (red), 53BP1+ cells (green; white arrowheads

indicate TIF) and nuclei stained with DAPI (blue)) in lung sections from *Trf1+/+* and *Trf1Δ/Δ* mice. Quantification of SCGB1A1+ (**b**), p63+ (**c**), and double SCGB1A1+-γH2AX+ (**d**), SCGB1A1+-p21+ (**f**), SCGB1A1+-Ki67+ (**g**) and SCGB1A1+-SOX2+ (**h**) club cells per epithelium length (mm), as well as the proportion (%) of SCGB1A1+ cells with more than 1 TIF (**e**) in *Trf1+/+* and *Trf1Δ/Δ* mice. Quantification of lung resistance (LR) (**i**) and dynamic compliance (Cdyn) (**j**) evaluated by plethysmography in *Trf1+/+* and *Trf1Δ/Δ* mice. Data are expressed as mean ± SEM (the number of mice is indicated in each case). *$p < 0.05$, **$p < 0.01$ (Mann–Whitney or unpaired *t* tests). Source data are provided as a Source Data file.

development onwards. For this purpose, we studied the *Trf1Δ/Δ p53−/− K5-Cre* mouse model[10]. The *Trf1Δ/Δ p53−/− K5-Cre* male mice exhibited a marked decreased survival compared to control counterparts (Supplementary Fig. 3b). Of note, immunostaining with K5 and TRF1 demonstrated that the majority of K5-positive cells stained negative for TRF1 (96%) in the lungs of *Trf1Δ/Δ* male mice (Supplementary Fig. 3c, d). In accordance with previous results (Fig. 10o–s), airway collagen (Sirius Red) and Vimentin stained areas, as well as airway SM thickness (SMA) were found significantly incremented in *Trf1Δ/Δ* male mice (Supplementary Fig. 3c, e, f, g).

## Discussion

In this study we have analyzed the lung pathological consequences of telomere dysfunction in fibroblasts, club and basal cells. We show that a short-term telomere dysfunction was efficiently induced in lung fibroblasts to study the early lung phenotypes, without affecting alveolar collagen deposition and fibroblast abundance, as previously shown in a similar mouse model[41]. Additionally, we show that upon telomere dysfunction COL1A2+ fibroblasts exhibited increased telomeric damage, cell cycle arrest, apoptosis and proliferation, as well as increased presence of inflammatory cells in BALF, but we did not observe decreased mouse survival unlike the study of Naikawadi et al. where telomere dysfunction in fibroblasts drastically decreased mouse survival 1 week after tamoxifen administration. These discrepancies could be possibly due to differences in the *Col1a2* Cre

driver, as well as to the dosing and timing of tamoxifen administration.

We also demonstrate that telomere dysfunction in fibroblasts in the context of a bleomycin (BLM)-induced fibrosis model, exacerbates classical features observed in patients with IPF including increased collagen deposition, fibroblast abundance and EMT, as well as reduced dynamic compliance[42–45]. Specifically, we previously demonstrated that a low BLM dose synergizes with short telomeres to trigger pulmonary fibrosis in telomerase-deficient mice[15]. Interestingly, after the BLM challenge, the majority of COL1A2+ cells stained positive for TRF1 in *Trf1Δ/Δ* mice, which could be due to the fact that fibroblasts that escaped the deletion of *Trf1* are those proliferating. Alternatively, an induction of EMT, indicated by the loss of E-cadherin[46], could be leading to a higher number of fibroblasts. Telomere dysfunction in lung fibroblasts exacerbated Th1 and Th2 inflammation upon BLM challenge. In this sense, M1 macrophages contribute to tissue injury after induction of Th1 inflammation and M2 macrophages lead to the resolution of inflammation and tissue repair upon activation of Th2 inflammation[47]. Specifically, M2 macrophages were reported to promote myofibroblast differentiation and are associated with pulmonary fibrogenesis[48].

In our study we also report that a telomere dysfunction was efficiently induced in club cells. We show that deletion of *Trf1* in club cells leads to a reduction in the number of these cells. As previously reported, *Trf1* deletion not only in club cells but also in lung fibroblasts and basal cells did not affect telomere length[10,11]. Unlike our study, telomere

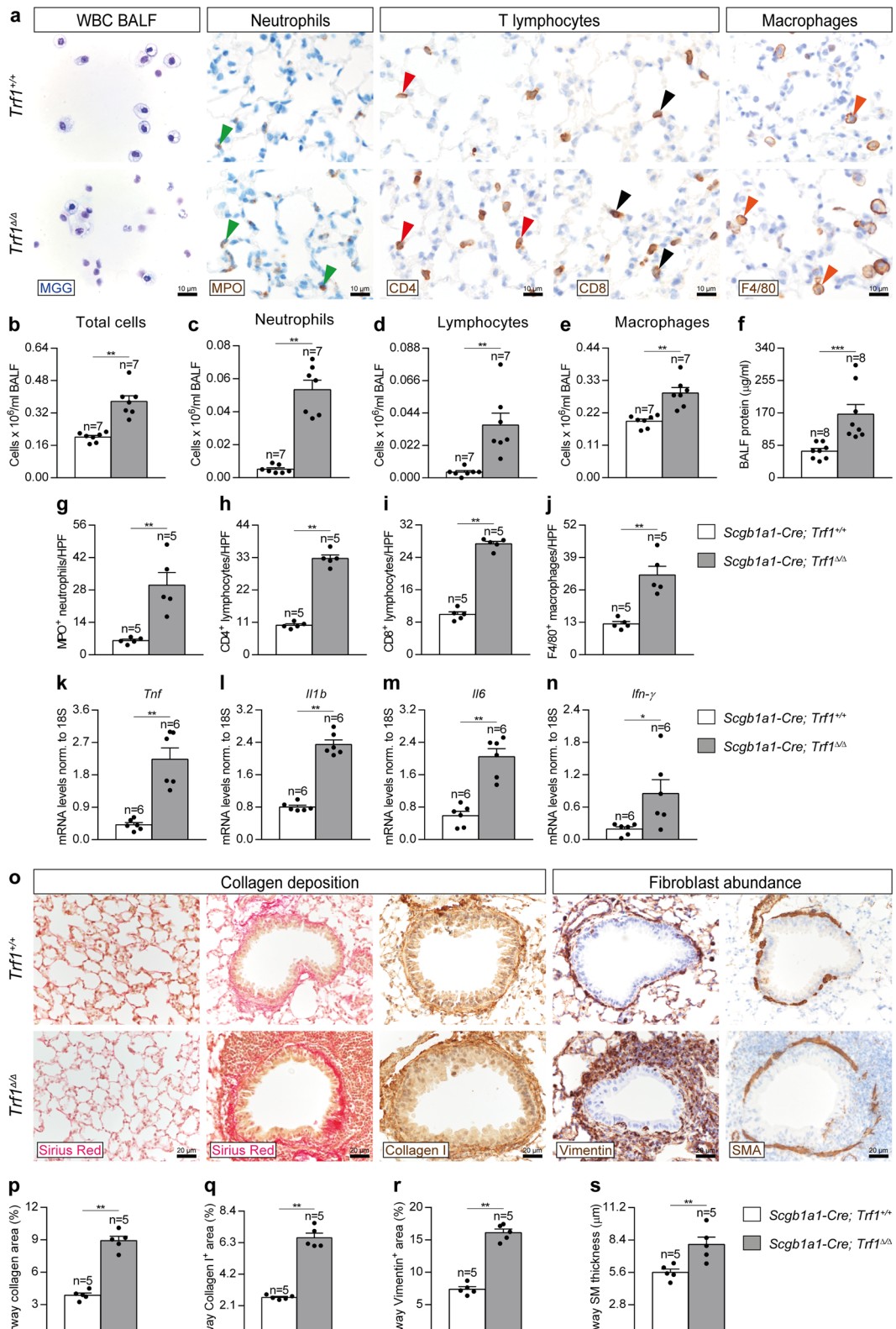

**Fig. 7 | *Trf1* deletion in club cells increases lung inflammation and airway remodeling. a** Representative BALF cytospin preparations (May-Grünwald Giemsa (MGG)) and immunostainings for MPO (neutrophils), CD4 and CD8 (T lymphocytes) and F4/80 (macrophages) in lung sections from *Trf1*⁺/⁺ and *Trf1*^Δ/Δ mice. Quantification of total (**b**) and differential BALF cell counts for neutrophils (**c**), lymphocytes (**d**) and macrophages (**e**), and total protein concentration in BALF (**f**) of *Trf1*⁺/⁺ and *Trf1*^Δ/Δ mice. Quantification of lung MPO (**g**), CD4 (**h**), CD8 (**i**) and F4/80 (**j**) positive cells per 40X high-power field (HPF), and lung tissue mRNA expression levels of *Tnf* (**k**), *Il1b* (**l**), *Il6* (**m**) and *Ifn-γ* (**n**) (Th1 inflammation) in *Trf1*⁺/⁺ and *Trf1*^Δ/Δ mice. **o** Representative stainings for Sirius Red (alveolar parenchyma and airways) and immunostainings for Collagen I, Vimentin and SMA (airways) in lung sections from *Trf1*⁺/⁺ and *Trf1*^Δ/Δ mice. Quantification of airway collagen (Sirius Red) (**p**), airway Collagen I (**q**) and Vimentin (**r**) positive areas (%), and airway smooth muscle (SM) thickness (SMA) (μm) (**s**) in *Trf1*⁺/⁺ and *Trf1*^Δ/Δ mice. Data are expressed as mean ± SEM (the number of mice is indicated in each case). *$p < 0.05$, **$p < 0.01$, ***$p < 0.001$ (Mann–Whitney or unpaired *t* tests). Source data are provided as a Source Data file.

length was reported to be reduced in mice with dysfunctional telomeres in club cells[49]. Furthermore, dysfunctional telomeres in club cells increased airway collagen content, inflammation and lung resistance (LR), in accordance to a previous work[49]. It should be noted that unlike our study, Naikawadi et al. did not assess fibroblast abundance, as well as telomeric damage, cell cycle arrest, proliferation and differentiation in SCGB1A1[+] club cells. Noteworthy, telomere dysfunction in club cells

did not increase alveolar collagen content, as we have reported in mice upon deletion of *Trf1* in ATII cells, which are at the origin of pulmonary fibrosis[15]. Specifically, we show increased differentiation of club cells upon deletion of *Trf1* in this cell type. Interestingly, we have recently reported in mice a significant increase of differentiating SOX2[+] club cells with aging, which was anticipated in telomerase-deficient mice[50]. SOX2 has been previously shown a marker of differentiation of club cells[51].

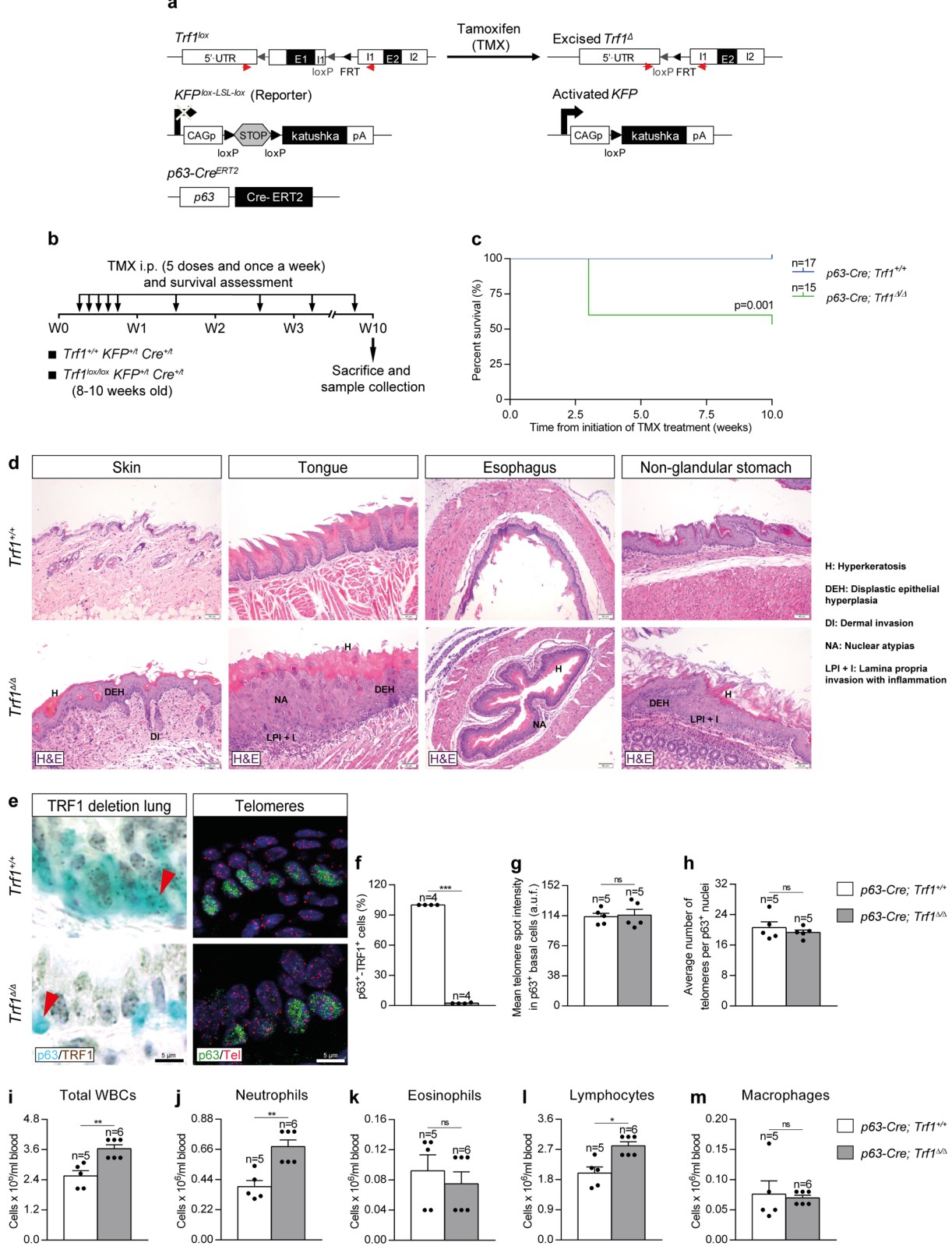

**Fig. 8 | Efficient *Trf1* deletion in lung basal cells upon tamoxifen administration. a** Generation of the conditional knockout mouse model in which *Trf1* was deleted in basal cells using the Cre recombinase driven by the *p63* promoter. *Trf1*[lox], *KFP*[Lox-LSL-Lox], and *Scgb1a1-Cre*[ERT2] alleles are depicted before and after Cre-mediated excision. **b** Tamoxifen (TMX) treatment, survival rate assessment and sample collection. Eight-to 10-week-old male *Trf1*[+/+] *KFP*[+/t] *Cre*[+/t] (*p63-Cre; Trf1*[+/+]) and *Trf1*[lox/lox] *KFP*[+/t] *Cre*[+/t] (*p63-Cre; Trf1*[lox/lox]) mice were i.p. injected with TMX for five consecutive days during the first week and once a week until the sacrifice and sample collection on week (W) 10. **c** Kaplan–Meier survival curves of *p63-Cre; Trf1*[+/+] (*Trf1*[+/+], controls) and *p63-Cre; Trf1*[Δ/Δ] (*Trf1*[Δ/Δ]) mice upon TMX treatment. **d** Abnormal pathologies observed in *Trf1*[Δ/Δ] mice in the skin (hyperkeratosis (H), dysplastic epithelial hyperplasia (DEH) and dermal invasion (DI)), tongue (hyperkeratosis (H), dysplastic epithelial hyperplasia (DEH), nuclear atypias (NA) and epithelial lamina propria invasion with inflammation (LPI + I)), esophagus (hyperkeratosis (H) and nuclear

atypias (NA)) and stomach (hyperkeratosis (H), dysplastic epithelial hyperplasia (DEH) and epithelial lamina propria invasion with inflammation (LPI + I)). **e** Representative immunostainings for p63 (blue) and TRF1 (brown) (red arrowheads indicate p63[+] basal cells with deletion of TRF1), and immune-telomere-Q-FISH in p63[+] club cells (Cy3Tel probe (red), p63[+] cells (green), and nuclei stained with DAPI (blue)) in lung sections from *Trf1*[+/+] and *Trf1*[Δ/Δ] mice. Quantification of p63[+]-TRF1[+] cells (%) (**f**), mean telomere spot intensity (**g**) and average number of telomeres (**h**) in SCGB1A1[+] cells in *Trf1*[+/+] and *Trf1*[Δ/Δ] mice. **i–m** Quantification of total white blood cells (**i**), neutrophils (**j**), eosinophils (**k**), lymphocytes (**l**) and macrophages (**m**) in peripheral blood from *Trf1*[+/+] and *Trf1*[Δ/Δ] mice. Data are expressed as mean ± SEM (the number of mice is indicated in each case). *$p < 0.05$, **$p < 0.01$, ***$p < 0.001$ (Mann–Whitney or unpaired *t* tests). Animal survival was assessed by the Kaplan–Meier analysis, using the log Rank (Mantel–Cox) test. Source data are provided as a Source Data file.

Concerning mouse survival, Naikawadi et al. reported that *Trf1*[Δ/Δ] mice started to die from 8 months upon TMX administration, unlike our study in which we show that *Trf1*[Δ/Δ] mice started to die from week 11, thus, we decided to study the early lung phenotypes. This discrepancy could be due to differences in the *Scgb1a1* Cre driver, as well as to the dosing and timing of tamoxifen administration. Interestingly, telomere dysfunction in club cells increased the number of p63[+] basal cells specifically in distal airways. In steady state, basal cells are quiescent and only present in the trachea and proximal intrapulmonary airways. Nevertheless, in response to lung injury, airway basal cells are activated to operate as progenitor cells capable of self-renewal and differentiation into Club cells[33,35,36,52,53].

Telomere dysfunction in p63[+] basal cells caused increased telomeric damage and cell cycle arrest, generating abnormal pathologies in the skin, tongue, esophagus and non-glandular stomach as we reported after deletion of *Trf1* in K5[+] basal cells[10], supporting the reduced mouse survival observed upon telomere dysfunction in both p63[+] and K5[+] basal cells. Thus, we had to study the early lung phenotypes since *Trf1*[Δ/Δ] mice started to die from week 3 upon TMX administration. *Trf1* deletion in p63[+] lung basal cells decreased the number of these cells but did not alter the number of club cells, which have been shown to be a stem cell population with an important role in lung repair[31,32]. We also show by the first time that dysfunctional telomeres in lung basal cells does not lead to fibrotic pathologies in the lung parenchyma but did increase lung inflammation, as well as airway collagen deposition, fibroblast abundance and lung resistance (LR). On this basis, we think that increased mortality of *Trf1*[Δ/Δ] mice is mainly due to intestinal defects originated by esophagus lumen reduction and stomach pathologies rather than lung dysfunction.

In summary, only male mice with dysfunctional telomeres in club and basal cells exhibited increased lung resistance (LR), airway collagen content and fibroblast abundance. This finding goes in line with the fact that androgens exacerbate lung function impairment and airway collagen deposition in bleomycin-challenged male versus female mice[54]. Indeed, IPF predominantly affects males[55]. Similarly, male mice are more prone than females to develop lung fibrosis[56]. Furthermore, telomere length, a risk factor for IPF[19], was described to be shorter in males than in females, since estrogen activates telomerase[57,58].

Specifically, dysfunctional telomeres in club and basal cells increased airway remodeling in male mice, which is a critical feature of several chronic bronchial diseases characterized by aberrant repair of the epithelium and accumulation of fibroblasts, which contribute to extracellular matrix deposition that involves the expression of mesenchymal proteins such as α-SMA and vimentin[59]. Airway remodeling is present in several bronchial diseases including asthma, chronic obstructive pulmonary disease (COPD), bronchiolitis obliterans (BO), bronchopulmonary dysplasia (BPD), cystic fibrosis and bronchiectasis[39,60–64]. Specifically, airway remodeling upon deletion of *Trf1* in club and basal cells is similar to that observed in asthmatic patients, mainly characterized by increased airway smooth muscle

mass and sub-epithelial fibrosis[39]. Of note short telomeres were reported in patients with bronchial diseases including asthma, COPD, BO, BPD[65–68]. In addition, dysfunctional telomeres were also observed in patients with COPD[69]. Our results do not ultimately probe that telomere dysfunction per se in club and basal cells lead to airway fibrosis since airway fibrosis could also be mediated by the inflammatory cascade triggered by TRF1 depletion.

Remarkably, the observed lung phenotypes in club and basal cells are induced by *Trf1*-dependent telomere dysfunction independently of telomere length. In particular, *Trf1* deletion induces telomere uncapping and the activation of a persistent DNA damage response (DDR) at chromosome ends[10,11].

On that basis we can conclude that TRF1 could have an important role in lung tissue homeostasis, supported by previous findings in which we demonstrated its importance in tissue regeneration and homeostasis upon conditional deletion of *Trf1* from specific cell types[10,13–15]. In summary, here we show the pathological consequences of telomere dysfunction in lung fibroblasts, club and basal cells. Specifically, telomere dysfunction in club and basal cells from male mice increased lung damage, inflammation and airway remodeling. Noteworthy, depletion of TRF1 in fibroblasts, Club and basal cells did not lead to interstitial lung fibrosis, underscoring ATII cells as the relevant cell type for the origin of interstitial fibrosis (Fig. 10t). Our findings contribute to a better understanding of the importance of TRF1 in lung tissue homeostasis.

## Methods

### Ethical statement

All experiments and animal procedures were approved by our Institutional Animal Care and Use Committee (IACUC) (IACUC.011-2018, CBA_20_2018), by the Ethics Committee for Research and Animal Welfare (CEIyBA) (CBA 20-2018) from the Instituto de Salud Carlos III, and by Consejería de Medio Ambiente, Administración Local y Ordenación del Territorio (Comunidad de Madrid) (PROEX 163/18). All experiments and animal procedures were performed in accordance with the guidelines stated in the International Guiding Principles for Biomedical Research Involving Animals, developed by the Council for International Organizations of Medical Sciences (CIOMS). The animals were bred and maintained under specific pathogen-free (SPF) conditions in laminar flow caging at the CNIO animal facility in accordance with the recommendations of the Federation of European Laboratory Animal Science Associations (FELASA).

### Generation of mutant mouse lines

*Trf1*[lox/lox] mice were generated as previously described[10]. To conditionally delete *Trf1* in fibroblasts, club and basal cells, homozygous *Trf1*[lox/lox] mice were crossed with transgenic mice expressing *Cre*[ERT2] under the control of the *Col1a2*, *Scgb1a1* and *p63* promoters[31,37,40] as well as with transgenic mice harboring the Katushka fluorescent protein (KFP) encoding gene that contains a stop cassette flanked by lox

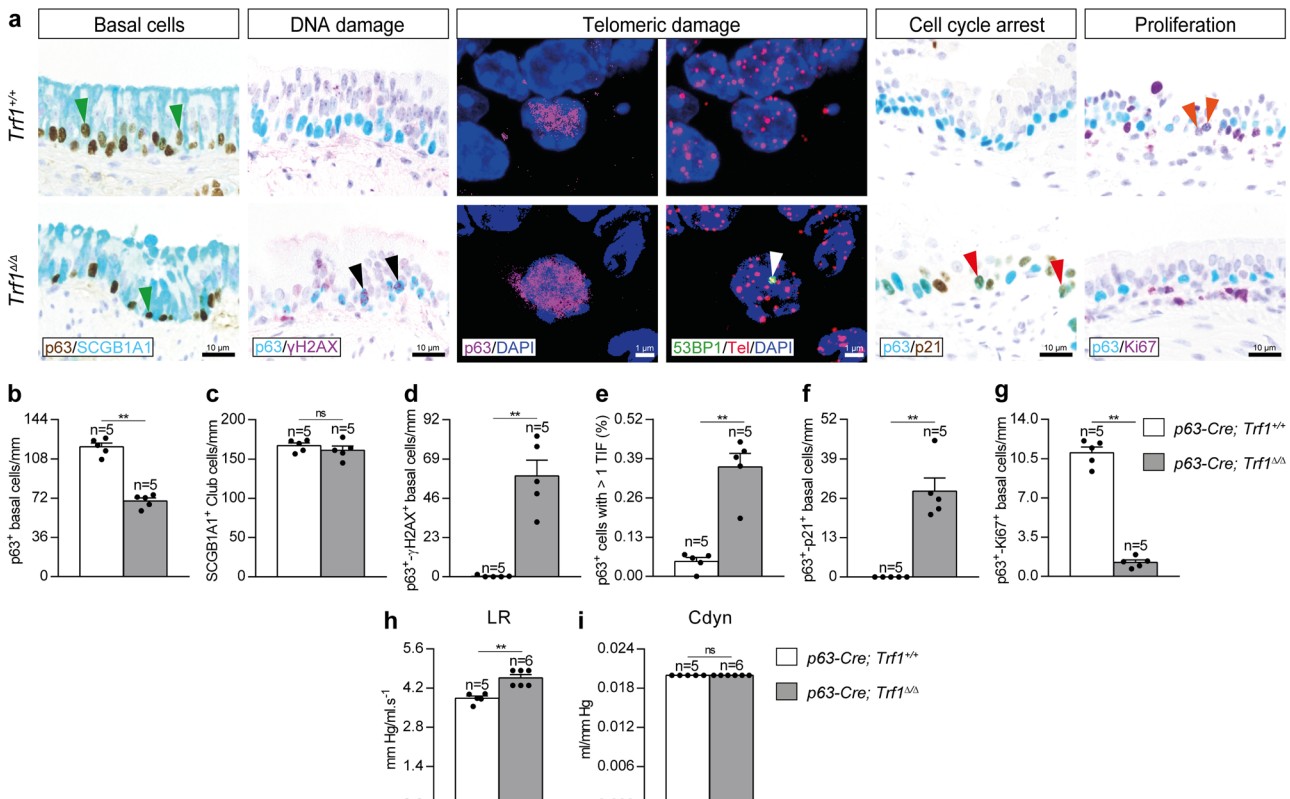

**Fig. 9 | *Trf1* deletion in basal cells increases telomeric damage and cell cycle arrest, and reduces proliferation of lung basal cells. a** Representative immunostainings for p63 (blue) and SCGB1A1 (brown; green arrowheads indicate p63⁺ basal cells), p63 (blue) and γH2AX (brown; black arrowheads indicate double p63⁺-γH2AX⁺ basal cells), p63 (blue) and p21 (brown; red arrowheads indicate double p63⁺-p21⁺ basal cells), and p63 (blue) and Ki67 (purple; orange arrowheads indicate double p63⁺-Ki67⁺ basal cells), as well as representative images of telomeric induced foci (TIF) in p63⁺ cells (p63 (purple), Cy3Tel probe (red), 53BP1⁺ cells (green; white arrowheads indicate TIF) and nuclei stained with DAPI (blue)) in lung

sections from *Trf1⁺/⁺* and *Trf1^Δ/Δ^* mice. Quantification of p63⁺ (**b**), SCGB1A1⁺ (**c**) and double p63⁺-γH2AX⁺ (**d**), p63⁺-p21⁺ (**f**) and p63⁺-Ki67⁺ (**g**) basal cells per epithelium length (mm), as well as the proportion (%) of p63⁺ cells with more than 1 TIF (**e**) in *Trf1⁺/⁺* and *Trf1^Δ/Δ^* mice. Quantification of lung resistance (LR) (**h**) and dynamic compliance (Cdyn) (**i**) evaluated by plethysmography in *Trf1⁺/⁺* and *Trf1^Δ/Δ^* mice. Data are expressed as mean ± SEM (the number of mice is indicated in each case). **$p < 0.01$ (Mann–Whitney or unpaired *t* tests). Source data are provided as a Source Data file.

sequences, the *KFP^CAG-lox-STOP-lox^* allele[70] (Figs. 1a, 5a and 8a). Tamoxifen (TMX) (Sigma Aldrich, San Luis, MO) was intraperitoneally (i.p.) injected to eight-to 10-week-old male *Trf1⁺/⁺ KFP⁺/t Cre⁺/t* and *Trf1^lox/lox^ KFP⁺/t Cre⁺/t* mice daily for five consecutive days during the first week and then once a week until the sacrifice and sample collection (Figs. 1b, 5b and 8b). Fluorescence intensity of KFP was measured with an IVIS Lumina Series III (Perkin Elmer, Waltham, MA) in vivo imaging system. Additionally, we have generated the *Trf1^Δ/Δ^ p53⁻/⁻ K5-Cre* mouse model, in which a *Trf1* deletion was induced from embryonic development onwards(Supplementary Fig. 3a)[10].

### Bleomycin mouse model
TMX (Sigma Aldrich) was i.p. injected to eight-to 10-week-old male *Trf1⁺/⁺ KFP⁺/t Cre⁺/t* (*Col1a2-Cre; Trf1⁺/⁺*) and *Trf1^lox/lox^ KFP⁺/t Cre⁺/t* (*Col1a2-Cre; Trf1^lox/lox^*) mice for five consecutive days during the first week and then once a week until week (W) 7 to induce the deletion of *Trf1* in *Col1a2⁺* fibroblasts (Fig. 3a). Then, at W7, animals were intra-tracheally instilled with either a single dose of 0.8 mg/kg of bleomycin (BLM) (Sigma Aldrich, San Luis, MO) or saline (controls) under a ketamine-medetomidine anesthetic combination (10 μl/g). Sacrifice and sample collection was performed at W10. (Fig. 3a).

### In vivo measurement of lung function
The mice were anesthetized using 10 μl/g of ketamine-medetomidine and intubated with a 24-gauge catheter (BD biosciences, Franklin lakes, NJ, USA). Then, lung function was assessed in a plethysmograph

(SCIREQ, Montreal, Canada) for the determination of LR (lung resistance) and Cdyn (dynamic compliance)[71,72].

### Sample collection and processing
Animals were euthanized using 10 μl/g of ketamine-xylazine. Blood was collected by cardiac puncture and lungs were lavaged with 1 ml of cold PBS 1X. Right lung lobes were dissected and snap-frozen in liquid nitrogen for qPCR and ELISA analyses, and the left lung lobe and remaining organs were fixed in 10% buffered formalin, embedded in paraffin and cut into 3 μm sections for histopathological evaluation, immunohistochemistry (IHC) or immunofluorescence (IFC).

### Histopathological analyses, IHC and IFC
Hematoxylin and eosin (H&E) staining was performed for histopathological evaluation, and Sirius Red (Sigma-Aldrich) staining served to evaluate collagen deposition. IFC was performed using the following antibodies: COL1A2 (Clone E-6 1:400, Santa Cruz Biotechnology, Dallas, TX) and TRF1 (CNIO Monoclonal Antibodies Core Unit, Madrid, Spain). IHC was performed using the following antibodies: Turbo-RFP (KFP) (1:3000, EVROGEN, Moscow, Russia), COL1A2 (Clone E-6 1:100, Santa Cruz Biotechnology), H2AX (Ser139, Clone JBW301 1:200, EMD Millipore, Burlington, MA), p16 (Clone 33B 1:30, CNIO Monoclonal Antibodies Core Unit), p21 (Clone 291H/B5 1:10, CNIO Monoclonal Antibodies Core Unit), C3 (Asp175 1:300, Cell Signaling Technology, Danvers, MA), Ki-67 (Clone D3B5 1:50, Cell Signaling Technology), SCGB1A1/CC10 (Clone T-18 1:1000, Santa Cruz Biotechnology), TRF1

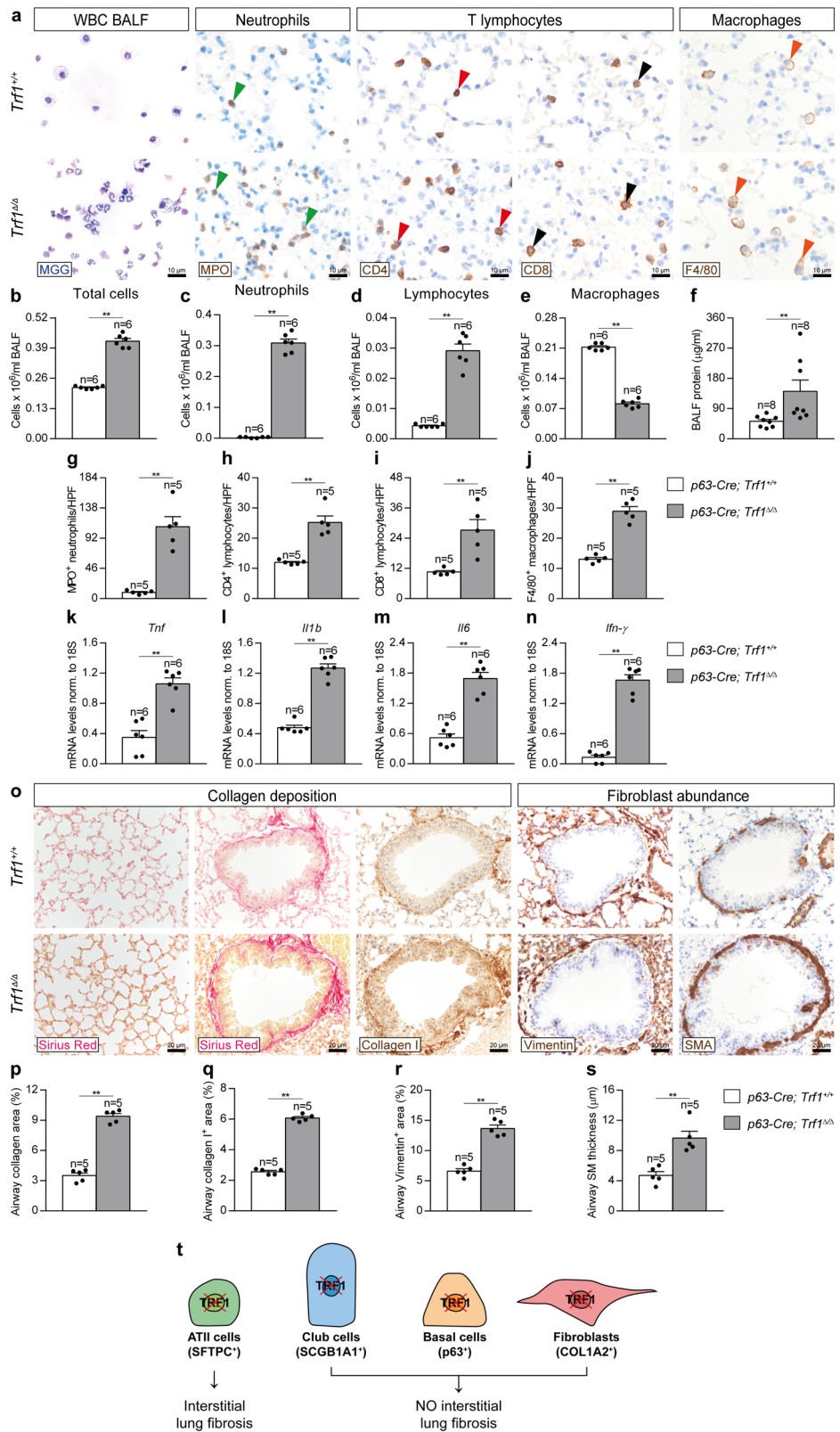

(Clone 572C 1:50, Abcam, Cambridge, UK), p63 (Clone 4A4, Roche, Basel, Switzerland), SOX2 (Clone C70B1 1:75, Cell Signaling Technology), Myeloperoxidase (1:1250, DAKO, Jena, Germany), CD4 (Clone D7D2Z 1:50, Cell Signaling Technology), CD8 (Clone 94A 1:200, CNIO Monoclonal Antibodies Core Unit), F4/80 (Clone A3-1 MCA497 1:20, AbD Serotec/Bio-Rad, Hercules, CA), Collagen I (1:600, EMD Millipore), Vimentin (Clone D21H3 1:50, Cell Signaling Technology), E-cadherin

(Clone 36 1:1000, BD Biosciences) and SMA (Clone 1A4 1:4, DAKO, Agilent technologies, Santa Clara, CA).

## Telomere Q-FISH analyses
After deparaffinization and rehydration, tissues underwent antigen retrieval in 10 mM sodium citrate buffer and permeabilization was performed in PBS 0.5% Triton X-100 for 3 h. Next, tissues were washed

**Fig. 10 | *Trf1* deletion in lung basal cells increases lung inflammation and airway remodeling. a** Representative BALF cytospin preparations (May-Grünwald Giemsa (MGG)) and immunostainings for MPO (neutrophils), CD4 and CD8 (T lymphocytes) and F4/80 (macrophages) in lung sections from *Trf1*[+/+] and *Trf1*[Δ/Δ] mice. Quantification of total (**b**) and differential BALF cell counts for neutrophils (**c**), lymphocytes (**d**) and macrophages (**e**), and total protein concentration in BALF (**f**) of *Trf1*[+/+] and *Trf1*[Δ/Δ] mice. Quantification of lung MPO (**g**), CD4 (**h**), CD8 (**i**) and F4/80 (**j**) positive cells per 40X high-power field (HPF), and lung tissue mRNA expression levels of *Tnf* (**k**), *Il1b* (**l**), *Il6* (**m**) and *Ifn-γ* (**n**) (Th1 inflammation) in *Trf1*[+/+] and *Trf1*[Δ/Δ] mice. **o** Representative stainings for Sirius Red (alveolar parenchyma and airways) and immunostainings for Collagen I, Vimentin and SMA (airways) in lung sections from *Trf1*[+/+] and *Trf1*[Δ/Δ] mice. **p–s** Quantification of airway collagen (Sirius Red), airway Collagen I and Vimentin positive areas (%), and airway smooth muscle (SM) thickness (SMA) (μm) in *Trf1*[+/+] and *Trf1*[Δ/Δ] mice. Data are expressed as mean ± SEM (the number of mice is indicated in each case). **$p < 0.01$ (Mann–Whitney or unpaired $t$ tests). **t** Pathological consequences of telomere dysfunction in fibroblasts, club and basal cells in the lung. Dysfunctional telomeres in alveolar type II (ATII) cells led to alveolar DNA damage, senescence and apoptosis, as well as to interstitial lung fibrosis[15]. TRF1 deficiency in Club and basal cells induced telomeric damage and cell cycle arrest, and reduced proliferation of these cell types. TRF1 deletion in fibroblasts increased telomeric damage, cell cycle arrest, apoptosis and proliferation in this cell type. Noteworthy, depletion of TRF1 in fibroblasts, Club and basal cells did not lead to interstitial lung fibrosis. Source data are provided as a Source Data file.

3 × 5 min in PBS 1X, fixed in 4% formaldehyde for 5 min, washed 3 × 5 min in PBS and dehydrated in a 70%–90%–100% ethanol series (5 min each). Then, the immuno-telomere-Q-FISH with COL1A2 (Clone E-6 1:400, Santa Cruz Biotechnology), SCGB1A1/CC10 (Clone E-11 1:100, Santa Cruz Biotechnology) and p63 (Clone 4A4, Roche) antibodies was performed and analyzed as previously described[50,71]. Following the same protocol, an immuno-telomere-Q-FISH with the DNA damage marker 53BP1 (1:500, Novus Biologicals, Centennial, CO) was performed to identify telomeric induced foci (TIF) in COL1A2 (Clone E-6 1:400, Santa Cruz Biotechnology), SCGB1A1 (Clone E-11 AF488 1:100, Santa Cruz Biotechnology) and p63 (Clone 4A4, Roche, Basel, Switzerland) positive cells as previously described[71,73].

### RNA isolation, reverse transcription, qPCR and ELISAS
Inferior right lung lobes were homogenized in TRIzol reagent (Invitrogen), and RNA was isolated using a RNeasy Mini Kit (Qiagen, Hilden, Germany) and reverse-transcribed to cDNA using SuperScript II First-Strand Synthesis System (Invitrogen). qPCR was performed as previously describe[50,74]. Primer sets used for qPCR are included within the supplementary information (Supplementary Table 1). TGFB1 levels were assessed in homogenized lung tissue lysates using a TGFB1 Quantikine ELISA Kit (R&D systems, Minneapolis, MN).

### Statistics
According to the sample distribution (Shapiro–Wilk normality test), either a Mann–Whitney or umpaired $t$ tests were used to compare differences between 2 independent groups. Following a Shapiro–Wilk normality test, either a one-way ANOVA test or a Kruskal–Wallis test were used and then, the post hoc Dunn–Sidak multiple test was carried out for multiple comparisons between experimental groups. Animal survival was assessed by the Kaplan–Meier analysis, using the log Rank (Mantel–Cox) test.

### Reporting summary
Further information on research design is available in the Nature Research Reporting Summary linked to this article.

## Data availability
The authors declare that data supporting the findings of this study are available within the paper (and its supplementary information files). Source data are provided with this paper.

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

## Acknowledgements

We are grateful to Dr. J. Xu from the Baylor College of Medicine for providing *p63-CreERT2* mouse sperm for the generation of the *p63* mutant mouse line. Research in the Blasco Lab is funded by AstraZeneca; Fundación Botín and Banco Santander (Spain); Agencia Estatal de Investigación (AEI/MCI/10.13039/501100011033) with the project RETOS SAF2017-82623-R, cofunded by European Regional Development Fund (ERDF), "A way of making Europe"; Comunidad de Madrid with the Synergy Project COVIDPREclinicalMODels-CM and the European Research Council (ERC) under the European Union's Horizon 2020 research and innovation programme (grant agreement No 882385) through the project ERC-AvG SHELTERINS. The CNIO, certified since 2011 as Severo Ochoa Centre of Excellence by AEI/MCI/10.13039/501100011033, is supported by the Spanish Government through the Instituto de Salud Carlos III (ISCIII).

## Author contributions

M.A.B. had the original idea and secured funding. M.A.B., P.M., J.C. and R.L. supervised research. M.A.B., P.M. and S.P.-H. wrote the paper. S.P.-H., G.B., J.M.F. and S.S. performed experiments. M.A.B., S.P.-H., G.B. and J.M.F. analyzed the data.

## Competing interests

The authors declare no competing interests.
