## [Peer Review File · Nature Communications]

Consequences of telomere dysfunction in fibroblasts, club and basal cells for lung fibrosis developmentEditorial Note: Parts of this Peer Review File have been redacted as indicated to maintain the confidentiality of unpublished data.

REVIEWER COMMENTS

Reviewer #1 (Remarks to the Author):

General Comments:

The authors previously published a manuscript showing that either: a) conditional TRF1 KO in ATII cells; or b) low dose bleomycin in germline telomerase deficient mice with short telomeres; leads to pulmonary inflammation and interstitial lung fibrosis (Cell Rep. 2015; PMID: 26146081). Here they extend their findings in the conditional TRF1 KO mouse model to the murine counterpart of 3 more (fibroblasts, club cells, and basal cells) of the 33 (15 epithelial, 9 endothelial, 9 stromal) molecularly defined parenchymal cell populations in the human lung (Nature, 2020; PMID: PMC7704697). The authors report 2 major findings. 1) Conditional TRF1 KO in fibroblasts does not cause lung pathology during the 12-week interval studied. 2) Conditional TRF1 KO in club cells or basal cells causes airway fibrosis – but strikingly, only in male mice.

I have 4 main concerns. 1) The authors chose not to explore the mechanism of their most important finding - the remarkable differences observed between males and females (inexplicably, this sex difference did not even make it into the Abstract and is repeatedly minimized in the Results and Discussion). 2. The authors draw conclusions about the role of telomere integrity in lung disease from a single model (in contrast to their prior publication using 2 models perturbing telomere function). 3. The amount of new information and the potential impact of the work is limited. As my comment above implies, there are many more lung cell populations that are yet to be studied in this manner, and each of them may teach us something about the importance of telomere integrity in lung homeostasis. However, this additional information may not be of interest to the general readership of Nature Communications, but more relevant to experts studying the details of lung biology. 4. It is not apparent to me that the authors have conclusively shown that telomere integrity per se can be interpreted as relevant to the human counterpart of the mouse lung pathologies shown (i.e., the airway fibrosis seen in severe persistent asthma). The authors have not excluded the possibility that the airway pathology observed could have resulted from any other exogenous or genetic perturbations causing airway inflammation and DNA damage – unrelated to telomere function. There are no experimental controls to address this possibility.

Specific Comments:

The timepoints and intervals of observation of each genetically modified mouse strain appears arbitrary. For each KO studied, the time interval of observation is different without any explanation: e.g., for fibroblasts, 12 weeks; for club cells, 22 weeks. Related to this concern, the Kaplan-Meier plots shown suggest the intervals of observation may have been insufficient to fully evaluate the survival impact of each KO intervention.

Fig 2: I cannot find the time point when pulmonary function studies (PFTs) were conducted. The legend goes from B to F, skipping C, D, E.

Fig 4, 7: Same problems. Missing time point for PFTs, missing legend letter subheadings.

Reviewer #2 (Remarks to the Author):

This study builds on a previous study from the Blasco lab demonstrating that Trf1 deletion in ATII cells in the lung is sufficient to induce pulmonary fibrosis. Here they examine the consequences of Trf1 deletion in fibroblasts, club and basal cells in transgenic mice. The authors report that loss of Trf1 in lung fibroblasts did not alter survival, telomere length, or lung function, and produced no evidence of pathology. Deletion of Trf1 in lung club cells also did not change survival or telomere length, but resulted in changes in male mice consistent with airway remodeling. Deletion of Trf1 in lung basal cells led to decreased survival, accompanied with skin alterations and pathologies. While there were no changes in telomere length, the authors observed changes consistent with airway remodeling in males. The manuscript appears to focus on the observations that Trf1 deletion in lung club and basal

cells, and fibroblasts, did not cause interstitial lung fibrosis, unlike Trf1 deletion in ATII cells (Figure 9). In other words, the manuscript focuses on a negative result. Based on this result the authors conclude that ATII are the relevant cell type for the origin of interstitial fibrosis. However, the physiological relevance for lung fibrosis in humans is unclear since this disease has been linked to critically short telomeres, rather than TRF1 loss. Would critically short telomeres in these cell types cause fibrosis? The result that Trf1 deletion in club and basal cells has a slight impact on lung function in male mice, as observed by a slight increase in lung resistance, is interesting and may warrant further study on the effects of lung function. Perhaps this is more relevant to other lung diseases that could be further developed. Specific comments are below.

1. Use of the KFP fluorescent protein to monitor Cre activity in vivo is a strength. The authors clearly show Trf1 expression is lost in the relevant cell types too. The methods and experiments are sound.

2. While the images in Fig 2A show an increase in gammaH2AX upon Trf1 deletion, it is not clear if these large foci colocalize with telomeres. The appearance of TIFs or TAFs would be strong evidence for telomere dysfunction in these cells. This lab is very experienced in staining for dysfunctional telomeres. The same comment applies to the other cell types in later figures.

3. In Figs. 2, 4, and 6 p21 staining alone is not a robust indicator of cellular senescence. Reduction in proliferation could be due to apoptosis as suggested. Since inflammatory markers are not significantly changed in fibroblasts (Fig. 2K-N) this suggests limited senescence induction, since an increase in SASP would be expected.

4. It is not clear why Trf1 deletion in lung club cells increased inflammation only in male mice, but not female mice. The observation that mRNA expression changes and immunohistological changes are consistent with airway remodeling upon Trf1 deletion in club and basal cells is interesting. The phenotypes appear to correlate with a slight increase in lung resistance in male mice only. Why might this be?

5. The authors conclude that ATII are the relevant cell type for the origin of interstitial fibrosis, because Trf1 deletion in fibroblasts, club and basal did not lead to lung fibrosis. However, the physiological relevance for lung fibrosis in humans is unclear since this disease has been linked to critically short telomeres, rather than TRF1 loss. Is the extent and level of telomere dysfunction comparable upon Trf1 deletion to that achieved with critically short telomeres?

Reviewer #3 (Remarks to the Author):

In this manuscript the authors investigate the effects of conditional deletion of Trf1 from lung fibroblasts, club cells and airway basal cells in mice. The work builds on prior published evidence that Trf1 deletion from alveolar type II cells causes telomere dysfunction and pulmonary fibrosis in mice. The major conclusion of the work is that Trf1 deletion in the cell populations studied here, unlike that observed after deletion in ATII cells, does not cause spontaneous pulmonary fibrosis in the alveoli. In general the work is thoroughly done, though there are some concerns with interpretation as detailed below. Additionally, the effects of club cell specific deletion of Trf1 are largely consistent with a recent publication that is mentioned but little discussed in the paper: <https://pubmed.ncbi.nlm.nih.gov/32551854/>, somewhat reducing the novelty.

Regarding the fibroblast-specific deletion of Trf1:

1. Why was the study halted at 12 weeks, when the club cell study was continued substantially

longer?

2. In Figure 2A, many of the Col1a2+ cells do not look anything like interstitial fibroblasts by their location, size or shape. There is a major concern here that the cells being quantified as fibroblasts are mislabeled, or that the deletion of Trf1 is activating Cre and reporter expression in additional cell populations. This might help to account for the unusual finding that fibroblasts are increasing apoptosis, senescence and proliferation at the same time. The authors must include more rigorous proof that the effects here are specific to fibroblasts.

3. Why is only p21 expression used as a marker of senescence? This seems to fall far short of definitive evidence of senescence.

4. It is perhaps not surprising that fibroblasts, which are largely quiescent in the healthy adult lung, are not triggering spontaneous fibrosis upon Trf1 deletion. How would the lung handle injury in this model of fibroblast specific telomere dysfunction? Some evaluation of the response to a fibrotic injury would seem essential to broaden the significance of this study.

5. A limitation of the model across all the tested cell types is the uniformity of the genetic deletion. In sporadic human disease there are likely to be more complex variations in telomere state. For fibroblasts (and perhaps the other cell types), the authors should consider how telomere dysfunction and/or senescence in a subpopulation of cells might change the results, and the overall interpretation of the findings.

For the club cell deletion studies:

6. It is striking how increased basal cells are in response to Trf1 deletion in club cells. Is this altered Scgb1a1 expression in proximal airways, or altered formation of basal cells in more distal airways? Some clarification and further mechanistic insight would increase the novelty of this section.

7. In panel O it is inappropriate to use SMA as a marker of fibroblast activation - these are smooth muscle cells that surround the airways and constitutively express SMA.

8. Vimentin is a poorly selective indicator of fibroblasts both here and elsewhere in the manuscript. Pdgfra and/or Col1a1/2 would be superior indicators.

For the basal cell studies:

9. Can the authors comment on the cause of increased mortality and whether it was related to the lung phenotype or other?

10. Again it is not clear why the relatively early time point for study conclusion was chosen.

11. Again SMA is a smooth muscle marker around the airways, and the staining shown cannot be interpreted as fibroblast activation.

12. It would be expected that the increased telomere dysfunction/senescence in basal cells would compromise the lung's response to injury. To conclude that basal cell telomere dysfunction is unrelated to distal lung fibrosis, the authors should compare the fibrosis and repair response of these mice to their wild type littermates.

DETAILED ANSWERS TO REVIEWERS

Detailed Answers to Reviewer #1:

General Comments:

The authors previously published a manuscript showing that either: a) conditional TRF1 KO in AII cells; or b) low dose bleomycin in germline telomerase deficient mice with short telomeres; leads to pulmonary inflammation and interstitial lung fibrosis (Cell Rep. 2015; PMID: 26146081). Here they extend their findings in the conditional TRF1 KO mouse model to the murine counterpart of 3 more (fibroblasts, club cells, and basal cells) of the 33 (15 epithelial, 9 endothelial, 9 stromal) molecularly defined parenchymal cell populations in the human lung (Nature, 2020; PMID: PMC7704697). The authors report 2 major findings. 1) Conditional TRF1 KO in fibroblasts does not cause lung pathology during the 12-week interval studied. 2) Conditional TRF1 KO in club cells or basal cells causes airway fibrosis – but strikingly, only in male mice.

I have 4 main concerns:

1) The authors chose not to explore the mechanism of their most important finding - the remarkable differences observed between males and females (inexplicably, this sex difference did not even make it into the Abstract and is repeatedly minimized in the Results and Discussion).

ANSWER: We thank the reviewer for the thorough review of our manuscript and for the commentaries and suggestions, which we have addressed in a revised manuscript. Concerning differences observed between males and females, we agree with the reviewer that it is an interesting finding that we now emphasize and discuss in the revised manuscript. In particular, only male mice with dysfunctional telomeres in club and basal cells exhibited increased lung resistance (LR), airway collagen content and fibroblast abundance. Interestingly, androgens were previously shown to exacerbate lung function impairment and airway collagen deposition in bleomycin-challenged male versus female mice (Voltz et al., *Am J Respir Cell Mol Biol* 2008, 39(1):45-52). Although several chronic lung diseases have a female predisposition, IPF predominantly affects males (Somayaji & Chalmers, *Eur Respir Rev* 2022, 31(163):210111). Similarly, male mice are described to be more prone than females to develop lung fibrosis (Kawano-Dourado et al., *European Respir Rev* 2021, 30(162):210105). Moreover, telomeres, a risk factor for IPF (Alder et al., *Proc Natl Acad Sci USA* 2008, 105(35):13051-13056), were proven to be shorter in males than in females (Gardner et al., *Exp Gerontol* 2014, 51:15-27), since estrogen activates telomerase (Bayne et al., *Cell Res* 2008, 18(11):1141-1150). We have included these facts in the revised manuscript (**pages 22, lines 483-491**).

2. The authors draw conclusions about the role of telomere integrity in lung disease from a single model (in contrast to their prior publication using 2 models perturbing telomere function).

ANSWER: In our study, we focused in *Trf1* deletion as we previously demonstrated that TRF1 deletion in Alveolar type II cells **is sufficient to lead to progressive and lethal** pulmonary fibrosis without the need of any additional damage to the lungs (Povedano et al., *Cell Rep* 2015, 12(2):286-299), and therefore comparison between different mouse strains in which TRF1 is deleted in different cell types will be informative on the cell of origin of fibrosis induced by telomere dysfunction. Nevertheless, in the revised manuscript, we now highlight that the observed lung phenotypes originate by TRF1-dependent telomere dysfunction and not but telomere length shortening (**pages 22-23, lines 506-510**).

3. The amount of new information and the potential impact of the work is limited. As my comment above implies, there are many more lung cell populations that are yet to be studied in this manner, and each of them may teach us something about the importance of telomere integrity in lung homeostasis. However, this additional information may not be of interest to the general readership of Nature Communications, but more relevant to experts studying the details of lung biology.

ANSWER: The reviewer is right that there are many other cell types within the cell lung population and that their impact in lung homeostasis is also of great interest. In this work, however, we have focused in the three major lung cell types, fibroblasts, club and basal cells. We hope the reviewer appreciates the enormous amount of work that involved generating three independent genetically modified mouse lines, and that the conclusions that we have generated are sound.

4. It is not apparent to me that the authors have conclusively shown that telomere integrity per se can be interpreted as relevant to the human counterpart of the mouse lung pathologies shown (i.e., the airway fibrosis seen in severe persistent asthma). The authors have not excluded the possibility that the airway pathology observed could have resulted from any other exogenous or genetic perturbations causing airway inflammation and DNA damage – unrelated to telomere function. There are no experimental controls to address this possibility.

ANSWER: We respectfully disagree with the reviewer in this point. We have included in all the experiments a mouse control group consisting of *Trf1*^{+/+} mice (wild-type) that is genetically similar and has been treated and handled similarly as the *Trf1*^{ΔΔ} experimental group, therefore ruling out phenotypes that are unrelated to telomere dysfunction. We fully agree with the reviewer that the lung phenotypes observed upon *Trf1* deletion are the consequences of inflammatory and DNA damage response induced by telomeric DNA damage.

Specific Comments

The timepoints and intervals of observation of each genetically modified mouse strain appear arbitrary. For each KO studied, the time interval of observation is different without any explanation: e.g., for fibroblasts, 12 weeks; for club cells, 22 weeks. Related to this concern, the Kaplan-Meier plots shown suggest the intervals of observation may have been insufficient to fully evaluate the survival impact of each KO intervention.

ANSWER: We understand the point of the reviewer on the different time points of each genetically modified mouse strain and that to perform long-term telomere dysfunction in these models would serve to better evaluate the survival impact. Concerning the *Col1a2* mutants, we decided to induce a short-term telomere dysfunction to study the early lung phenotypes. As we did not see significant lung pathologies in *Col1a2* mutants upon *Trf1* deletion, we now included in the revised manuscript the effect of TRF1 deficiency in fibroblasts in the context of a bleomycin-induced fibrosis model (**new Fig. 3** and **new Fig. 4**). Regarding the *Scgb1a1* mutants, Naikawadi *et al.* (Naikawadi *et al.*, *Am J Respir Cell Mol Biol* 2020, 63(4):490-501) reported that animals started to die from 8 months upon TMX administration. In our case, *Scgb1a1* mutants with telomere dysfunction started to die from week 11 upon TMX treatment, thus, we decided to study the lung phenotypes earlier than in the study reported by Naikawadi *et al.* Concerning *p63* mutants, we had to study the early lung phenotypes since these mutants started to die from week 3 upon TMX administration. We have discussed these issues in the revised version manuscript (**pages 19-21, lines 409-482**).

Fig 2: I cannot find the time point when pulmonary function studies (PFTs) were conducted. The legend goes from B to F, skipping C, D, E.

ANSWER: In the **Fig. 1b** and legend we had indicated that sacrifice and sample collection had been performed on week 7 upon TMX treatment (also for PFTs). We have also included the missing legend letter subheadings in the legend of **Fig. 2**.

Fig 4, 7: Same problems. Missing time point for PFTs, missing legend letter subheadings.

ANSWER: Concerning this issue, in **new Fig. 5b**, **new Fig.8b** and corresponding legends we have indicated the time point when animals were sacrificed and sample collection was performed (*Scgb1a1* mutants: 22 weeks upon initiation of TMX treatment; *p63* mutants: 10 weeks upon initiation of TMX administration). Moreover we have included the missing legend letter subheadings in the legends of **new Fig. 6** and **new Fig. 9**.

Detailed Answers to Reviewer #2

This study builds on a previous study from the Blasco lab demonstrating that *Trf1* deletion in ATII cells in the lung is sufficient to induce pulmonary fibrosis. Here they examine the consequences of *Trf1* deletion in fibroblasts, club and basal cells in transgenic mice. The authors report that loss of *Trf1* in lung fibroblasts did not alter survival, telomere length, or lung function, and produced no evidence of pathology. Deletion of *Trf1* in lung club cells also did not change survival or telomere length, but resulted in changes in male mice consistent with airway remodeling. Deletion of *Trf1* in lung basal cells led to decreased survival, accompanied with skin alterations and pathologies. While there were no changes in telomere length, the authors observed changes consistent with airway remodeling in males. The manuscript appears to focus on the observations that *Trf1* deletion in lung club and basal cells, and fibroblasts, did not cause interstitial lung fibrosis, unlike *Trf1* deletion in ATII cells (Figure 9). In other words, the manuscript focuses on a negative result. Based on this result the authors conclude that ATII are the relevant cell type for the origin of interstitial fibrosis. However, the physiological relevance for lung fibrosis in humans is unclear since this disease has been linked to critically short telomeres, rather than TRF1 loss. Would critically short telomeres in these cell types cause fibrosis? The result that *Trf1* deletion in club and basal cells has a slight impact on lung function in male mice, as observed by a slight increase in lung resistance, is interesting and may warrant further study on the effects of lung function. Perhaps this is more relevant to other lung diseases that could be further developed. Specific comments are below.

1. Use of the KFP fluorescent protein to monitor Cre activity in vivo is strength. The authors clearly show *Trf1* expression is lost in the relevant cell types too. The methods and experiments are sound.

ANSWER: We appreciate reviewer's opinion about the soundness of our work. We also thank the reviewer for the commentaries and suggestions, which we have addressed in the revised manuscript.

2. While the images in Fig 2A show an increase in gammaH2AX upon *Trf1* deletion, it is not clear if these large foci colocalize with telomeres. The appearance of TIFs or TAFs would be strong evidence for telomere dysfunction in these cells. This lab is very experienced in staining for dysfunctional telomeres. The same comment applies to the other cell types in later figures.

ANSWER: We agree with the reviewer in that the appearance of TIFs would be a strong evidence for telomere dysfunction in these cells. Therefore, in the revised manuscript, we have now analyzed the presence of telomeric induced foci (TIF) in COL1A2⁺ fibroblasts (**new Fig. 2a, c**), in SCGB1A1⁺ club cells (**new Fig. 6a, e**) and in p63⁺ basal cells (**new Fig 9a, e**) upon *Trf1* deletion. Importantly, we observed increased telomeric damage in fibroblasts, club and basal cells.

3. In Figs. 2, 4, and 6 p21 staining alone is not a robust indicator of cellular senescence. Reduction in proliferation could be due to apoptosis as suggested. Since inflammatory markers are not significantly changed in fibroblasts (Fig. 2K-N) this suggest limited senescence induction, since and increase in SASP would be expected.

ANSWER: We appreciate reviewer's comment on potential senescence induction in our genetically modified mouse strains. In this regard, we have performed double immunostainings for COL1A2, SCGB1A1 and p63 with the senescence markers p16 and p19 upon *Trf1* deletion in fibroblasts (**new Fig. 2a,d**), club and basal cells (**see Figure 1 for reviewers**). In summary, we only observed increased presence of double COL1A2⁺-p16⁺ fibroblasts (**Fig. 2a, d**). In the case of club and basal cells we did not detect double SCGBA1⁺-p16/p19 club cells or p63⁺-p16/p19 basal cells (**see Figure 1 for reviewers**). On this basis, we decided to replace the term "senescence" by "cell cycle arrest" in **Fig. 2a, Fig. 6a** and **Fig. 9a**, which is more appropriate in view of our results.

[redacted]

4. It is not clear why Trf1 deletion in lung club cells increased inflammation only in male mice, but not female mice. The observation that mRNA expression changes and immunohistological changes are consistent with airway remodeling upon Trf1 deletion in club and basal cells is interesting. The phenotypes appear to correlate with a slight increase in lung resistance in male mice only. Why might this be?

ANSWER: We agree with the reviewer that the difference between males and females is very intriguing. Concerning differences observed between males and females, we have emphasized this sex-disparity throughout the revised manuscript (page 22, lines 513-521). In particular, only male mice with dysfunctional telomeres in club and basal cells exhibited increased lung resistance (LR), airway collagen content and fibroblast abundance. In this regard, androgens are shown to exacerbate lung function impairment and airway collagen deposition in bleomycin-challenged male versus female mice (*Voltz et al., Am J Respir Cell Mol Biol 2008, 39(1):45-52*). Although several chronic lung diseases have a female predisposition, IPF predominantly affects males (*Somayaji & Chalmers, Eur Respir Rev 2022, 31(163):210111*). Similarly, male mice are more prone than females to develop lung fibrosis (*Kawano-Dourado et al., European Respir Rev 2021, 30(162):210105*). Moreover, telomeres, a risk factor for IPF (*Alder et al., Proc Natl Acad Sci USA 2008, 105(35):13051-13056*), were proven to be shorter in males than in females (*Gardner et al., Exp Gerontol 2014, 51:15-27*), since estrogen activates telomerase (*Bayne et al., Cell Res 2008, 18(11):1141-1150*). We have discussed these issues in the revised manuscript (**page 22, lines 483-491**).

5. The authors conclude that ATII are the relevant cell type for the origin of interstitial fibrosis, because Trf1 deletion in fibroblasts, club and basal did not lead to lung fibrosis. However, the physiological relevance for lung fibrosis in humans is unclear since this disease has been linked to critically short telomeres, rather than TRF1 loss. Is the extent and level of telomere dysfunction comparable upon Trf1 deletion to that achieved with critically short telomeres?

ANSWER: In the current study, we focused in *Trf1* deletion, as we previously demonstrated that TRF1 deletion in Alveolar type II cells is sufficient to lead to progressive and lethal pulmonary fibrosis without any additional damage to the lungs (*Povedano et al., Cell Rep 2015, 12(2):286-299*), and therefore comparison between different mouse strains in which TRF1 is deleted in different cell types would be very informative on the cell of origin of fibrosis induced by telomere dysfunction. Nevertheless, in the revised manuscript, we now highlight that the observed lung phenotypes originate by TRF1-dependent telomere dysfunction (**pages 22-23, lines 506-510**).

Detailed Answers to Reviewer #3:

In this manuscript the authors investigate the effects of conditional deletion of Trf1 from lung fibroblasts, club cells and airway basal cells in mice. The work builds on prior published evidence that Trf1 deletion from alveolar type II cells causes telomere dysfunction and pulmonary fibrosis in mice. The major conclusion of the work is that Trf1 deletion in the cell populations studied here, unlike that observed after deletion in ATII cells, does not cause spontaneous pulmonary fibrosis in the alveoli. In general the work is thoroughly done, though there are some concerns with interpretation as detailed below. Additionally, the effects of club cell specific deletion of Trf1 are largely consistent with a recent publication that is mentioned but little discussed in the paper: <https://pubmed.ncbi.nlm.nih.gov/32551854/>, somewhat reducing the novelty.

ANSWER: We appreciate that the reviewer considers **this work “is thoroughly performed”**, and we thank her/him for the very insightful commentaries and suggestions, which we have addressed in full in the revised manuscript.

Concerning the publication mentioned by the reviewer, Naikawadi *et al.* studied the lung phenotypes upon deletion of *Trf1* in Club cells in a long-term way. Specifically, Naikawadi *et al.* showed that animals with a telomere dysfunction in club cells started to die from 8 months upon TMX treatment. In our case, some *Scgb1a1* mutants started to die from week 11, thus, we decided to study the lung phenotypes earlier than in the study reported by Naikawadi *et al.* These discrepancies could be due to differences in the *Scgb1a1* Cre driver, as well as to the dosing and timing of tamoxifen administration. We have discussed these issues in the revised version of the manuscript (**pages 20-21, lines 456-461**).

Regarding the fibroblast-specific deletion of Trf1

1. Why was the study halted at 12 weeks, when the club cell study was continued substantially longer?

ANSWER: We agree with the reviewer in that to perform long-term telomere dysfunction in these models would serve to better evaluate the survival impact. Concerning the *Col1a2* mutants, we decided to induce a short-term telomere dysfunction to study the early lung phenotypes. As we did not see significant lung pathologies in *Col1a2* mutants upon *Trf1* deletion, we have now included in the revised manuscript the study of the effect of TRF1 deficiency in fibroblasts in the context of a bleomycin-induced fibrosis model (**new Fig. 3 and new Fig. 4**). Regarding the *Scgb1a1* mutants, Naikawadi *et al.* (Naikawadi *et al.*, *Am J Respir Cell Mol Biol* 2020, 63(4):490-501) reported that animals started to die from 8 months upon TMX administration. In our case, *Scgb1a1* mutants with telomere dysfunction started to die from week 11 upon TMX treatment, thus, we decided to study the lung phenotypes earlier than in the study reported by Naikawadi *et al.* Concerning p63 mutants, we had to study the early lung phenotypes since these mutants started to die from week 3 upon TMX administration. We have discussed these issues in the revised version manuscript (**pages 19-21, lines 409-482**).

2. In Figure 2A, many of the Col1a2+ cells do not look anything like interstitial fibroblasts by their location, size or shape. There is a major concern here that the cells being quantified as fibroblasts are mislabeled, or that the deletion of Trf1 is activating Cre and reporter expression in additional cell populations. This might help to account for the unusual finding that fibroblasts are increasing apoptosis, senescence and proliferation at the same time. The authors must include more rigorous proof that the effects here are specific to fibroblasts.

ANSWER: We respectfully disagree with the reviewer. All the stainings and quantifications in Fig. 2a-g (γ H2AX, TIFs, p16, p21, Caspase 3 and Ki67) were performed in combination with COL1A2 immunodetection, a specific marker for fibroblasts [Zheng *et al.*, *Am J Pathol* 2002, 160(5):1609-1617; Lee *et al.*, *Nat Commun* 2020, 11(1):4254]. To proof that, we have performed double immunofluorescence stainings of COL1A2 (fibroblasts, red) with IBA1 (macrophages, green) in lung sections from *Col1a2-Cre; Trf1 $\Delta\Delta$* mice to demonstrate that although the shape of COL1A2+ fibroblasts is similar to IBA1+ alveolar macrophages, COL1A2 is not expressed in alveolar macrophages (**Figure 2 for reviewers**).

[redacted]

3. Why is only p21 expression used as a marker of senescence? This seems to fall far short of definitive evidence of senescence.

ANSWER: We have now performed double immunostainings for COL1A2, SCGB1A1 and p63 with the senescence markers p16 and p19 upon *Trf1* deletion in fibroblasts (**New Fig. 2a, d**), club and basal cells (**Figure 1 for reviewers**). We only observed increased presence of double COL1A2⁺-p16⁺ fibroblasts (**new Fig. 2a, d**). In the case of club and basal cells we did not detect double SCGBA1⁺-p16/p19 club cells or p63⁺-p16/p19 basal cells (**Figure 1 for reviewers**). Based on these observations, we have replaced the term "senescence" by "cell cycle arrest" in **Fig. 2a, Fig. 6a** and **Fig. 9a**, which is more appropriate in view of our results.

[redacted]

4. It is perhaps not surprising that fibroblasts, which are largely quiescent in the healthy adult lung, are not triggering spontaneous fibrosis upon *Trf1* deletion. How would the lung handle injury in this model of fibroblast specific telomere dysfunction? Some evaluation of the response to a fibrotic injury would seem essential to broaden the significance of this study.

ANSWER: We agree with the reviewer in that evaluation of the response to a fibrotic injury would strengthen the results of our study. Therefore, as we did not see significant lung pathologies upon deletion of *Trf1* in lung fibroblasts, in the revised manuscript we now include the study of the effect of TRF1 deficiency in fibroblasts in the context of a bleomycin (BLM)-induced fibrosis model (**new Fig. 3** and **new Fig. 4**). We observed that dysfunctional telomeres in fibroblasts exacerbated profibrotic pathologies (**new Fig. 3**) and inflammation (**new Fig. 4**) upon BLM-induced pulmonary fibrosis.

5. A limitation of the model across all the tested cell types is the uniformity of the genetic deletion. In sporadic human disease there are likely to be more complex variations in telomere state. For fibroblasts (and perhaps the other cell types), the authors should consider how telomere dysfunction and/or senescence in a subpopulation of cells might change the results, and the overall interpretation of the findings.

ANSWER: We agree with the reviewer in that partial *Trf1* deletion in the cell type under study might have a different outcome. In this work, we used genetic models to specifically determine

the effects of telomere dysfunction in a particular cell type. Future work in which lower doses or shorter times of TMX administration might shed light on the lung pathological effects of partial TRF1 depletion. However, these studies would constitute an independent piece of work.

For the club cell deletion studies

6. It is striking how increased basal cells are in response to Trf1 deletion in club cells. Is this altered Scgb1a1 expression in proximal airways, or altered formation of basal cells in more distal airways? Some clarification and further mechanistic insight would increase the novelty of this section.

ANSWER: We would like to point out that telomere dysfunction in Club cells increased the number of p63⁺ basal cells specifically in distal airways. In steady state, basal cells are quiescent and are only present in the trachea and proximal intrapulmonary airways. Nevertheless, in response to lung airway injury basal cells were reported to be activated to operate as stem/progenitor cells capable of self-renewal and differentiation into Club cells, thus basal cells where considered essential for lung regeneration (*Zuo et al., Nature 2015, 517(7536):616-620*; *Shaykhiev, Eur Respir J. 2015, 46(4):894-897*; *Yang et al., Dev Cell 2018, 44(6):752-761*; *Dean & Snelgrove, Am J Respir Crit Care Med 2018, 198(11):1355-1356*; *Morrisey, Dev Cell 2018, 44(6):653-654*). We have discussed this issue in the revised version of the manuscript (**page 21, lines 461-466**).

7. In panel O it is inappropriate to use SMA as a marker of fibroblast activation - these are smooth muscle cells that surround the airways and constitutively express SMA.

ANSWER: Regarding the use of SMA as a marker of “fibroblast activation”, we have replaced this term in the manuscript by “fibroblast abundance” to indicate that SMA was used as a marker of fibroblast abundance throughout the revised manuscript.

8. Vimentin is a poorly selective indicator of fibroblasts both here and elsewhere in the manuscript. Pdgfra and/or Col1a1/2 would be superior indicators.

ANSWER: It should be noted that we use several markers of fibroblasts and collagen fibers through our analyses (Sirius red, Collagen I, Vimentin and SMA). Even though Vimentin is not a selective marker of fibroblasts, it was reported to be an indicator of fibroblast abundance [JCI Insight. 2019 Apr 4; 4(7): e123253]. Thus, Vimentin stainings reinforce the study of fibroblast abundance assessed by SMA stainings in our experimental groups. We have also replaced the term “fibroblast presence” by “fibroblast abundance” in the manuscript to indicate that Vimentin was used as a marker of fibroblast abundance.

For the basal cell studies

9. Can the authors comment on the cause of increased mortality and whether it was related to the lung phenotype or other?

ANSWER: In the revised manuscript, we now show epithelial pathologies resulting from *Trf1* deletion in other locations than the lung; skin, tongue, esophagus and non-glandular stomach (**new Fig. 8d**). We think that the increased mortality in this mouse model is due to intestinal defects originated by esophagus lumen reduction and stomach pathologies rather than lung dysfunction. We now discuss this issue in the revised manuscript (**page 21, lines 467-482**).

10. Again it is not clear why the relatively early time point for study conclusion was chosen.

ANSWER: We had to study the early lung phenotypes in p63 mutants since they started to die from week 3 upon TMX administration. As mentioned above, we think that the increased mortality was due to intestinal defects originated by esophagus lumen reduction and stomach pathologies.

11. Again SMA is a smooth muscle marker around the airways, and the staining shown cannot be interpreted as fibroblast activation.

ANSWER: We have removed the fibroblast activation statement in the revised version of the manuscript and instead we indicate that SMA staining was used as a marker of fibroblast abundance.

12. It would be expected that the increased telomere dysfunction/senescence in basal cells would compromise the lung's response to injury. To conclude that basal cell telomere dysfunction is unrelated to distal lung fibrosis, the authors should compare the fibrosis and repair response of these mice to their wild type littermates.

ANSWER: We agree with the reviewer in that increased telomere dysfunction/senescence in basal cells could compromise the response to lung injury. Unfortunately, we cannot perform such study in our p63 and K5 mutant mouse lines, since p63 mutants were in a poor condition only 1 week upon TMX administration, as well as K5 mutants, in which *Trf1* was deleted in basal cells from day E 11.5 of embryonic development onwards. Thus, the humanitarian endpoint had to be applied in these mice. In this respect, we asked for permission to our Ethics Committee for Research and Animal Welfare (CElyBA) to perform a bleomycin challenge, but they did not allow us to perform this challenge due to ethical issues regarding the poor condition of p63 and K5 mutants.

REVIEWER COMMENTS

Reviewer #1 (Remarks to the Author):

The authors have satisfactorily addressed each of my concerns except point 4 related to airway fibrosis. The logic underlying the response to point 4 is faulty. Indeed, as the authors assert, the experiment proves that the genotype $Trf1\Delta/\Delta$ is responsible for the observed airway fibrosis. However, it does not prove that telomere dysfunction per se caused the fibrosis as implied in the manuscript. There may be several steps between genotype and airway fibrosis mediated by the inflammatory cascade triggered in the $Trf1\Delta/\Delta$ mice and not in the wild type mice. This point should be clarified before publication.

Reviewer #2 (Remarks to the Author):

The authors have satisfied my previous concerns, primarily regarding experimental controls. The relevance of $Trf1$ deletion in mice to pulmonary fibrosis in humans that arises from critically short telomere may still be limited. However, the author note that both shortening and $Trf1$ loss cause telomere dysfunction.

Reviewer #3 (Remarks to the Author):

The authors have added substantial new data in Figures 3 and 4 and have largely addressed my critiques through revisions and additions to the text. The resulting revised manuscript provides a thorough analysis of $Trf1$ deletion effects in three major lung cell populations at the molecular, cellular and tissue levels. I have two remaining requests:

1. Given the substantial new data showing (quite strikingly) that $Trf1$ deletion in fibroblasts augments bleomycin-induced fibrosis, this result should be included in the abstract.
2. The authors have corrected the usage of α SMA in the main figures, but Supplemental Figures 1-3 still label α SMA as "fibroblast activation" and this should also be corrected, as it largely labels the airway smooth muscle surrounding the airways.

DETAILED ANSWERS TO REVIEWERS

Detailed Answers to Reviewer #1:

Reviewer #1 (Remarks to the Author):

The authors have satisfactorily addressed each of my concerns except point 4 related to airway fibrosis.

We thank the reviewer for considering we have addressed almost all of his/her concerns.

The logic underlying the response to point 4 is faulty. Indeed, as the authors assert, the experiment proves that the genotype $Trf1^{\Delta/\Delta}$ is responsible for the observed airway fibrosis. However, it does not prove that telomere dysfunction per se caused the fibrosis as implied in the manuscript. There may be several steps between genotype and airway fibrosis mediated by the inflammatory cascade triggered in the $Trf1^{\Delta/\Delta}$ mice and not in the wild type mice. This point should be clarified before publication.

ANSWER: We have added a sentence in the Discussion (**pages 22-23, lane 508-511**) clarifying this concern: *“Our results do not ultimately probe that telomere dysfunction per se in club and basal cells lead to airway fibrosis since airway fibrosis could also be mediated by the inflammatory cascade triggered by TRF1 depletion”*.

Detailed Answers to Reviewer #2:

Reviewer #2 (Remarks to the Author):

The authors have satisfied my previous concerns, primarily regarding experimental controls. The relevance of $Trf1$ deletion in mice to pulmonary fibrosis in humans that arises from critically short telomere may still be limited. However, the author notes that both shortening and $Trf1$ loss cause telomere dysfunction.

We thank the reviewer for considering we have satisfied his/her concerns.

Detailed Answers to Reviewer #2:

Reviewer #3 (Remarks to the Author):

The authors have added substantial new data in Figures 3 and 4 and have largely addressed my critiques through revisions and additions to the text. The resulting revised manuscript provides a thorough analysis of $Trf1$ deletion effects in three major lung cell populations at the molecular, cellular and tissue levels.

We thank the reviewer for considering we have largely addressed his/her critiques and appreciate reviewer's opinion about the thorough analysis of $Trf1$ deletion effects in the three major lung cell populations at the molecular, cellular and tissue levels.

I have two remaining requests:

1. Given the substantial new data showing (quite strikingly) that Trf1 deletion in fibroblasts augments bleomycin-induced fibrosis, this result should be included in the abstract.

ANSWER: We have included in the abstract “*While Trf1 deletion in fibroblasts does not spontaneously lead to significant lung pathologies, upon bleomycin challenge TRF1 deficiency in fibroblasts exacerbates the inflammatory response and lung fibrosis*”.

2. The authors have corrected the usage of aSMA in the main figures, but Supplemental Figures 1-3 still label aSMA as "fibroblast activation" and this should also be corrected, as it largely labels the airway smooth muscle surrounding the airways.

ANSWER: We have corrected the labelling for a-SMA in Figures S1-S3 that now reads as “*Fibroblast abundance*”.